# Roles of the zona pellucida in gamete fusion and of the perivitelline space in blocking polyspermy in mice

Yaëlle Dubois [1], Sophie Favier[1,2,5], Nathan Martin-Fornier[1,5], Adrien Freyss[1], Mohyeddine Omrane[1], David Stroebel [3], Eric Perez [1], Sandrine Barbaux [2], Ahmed Ziyyat [2,4], Nicolas Rodriguez [1✉] & Christine Gourier [1✉]

## Abstract

The mechanisms regulating gamete fusion and preventing polyspermy in mammalian fertilization remain incompletely understood. This study combines real-time imaging, confocal microscopy and statistical analysis to investigate fertilization and polyspermy prevention dynamics in mice. By tracking the behavior of over one hundred spermatozoa entering the perivitelline space of oocytes, we dissect the respective contributions of oocyte structures (zona pellucida (ZP), perivitelline space (PVS), oolemma) and sperm components (head, flagellum) to fertilization and polyspermy prevention. We find that fertilization requires specific sperm head movements on the oolemma, driven by flagellar beating and facilitated by trapping the flagellum in the ZP, revealing a novel role for this structure. Our kinetic analysis characterizes a slow "penetration block" that gradually limits sperm entry into the PVS and a faster "fusion block" that prevents further fusion events. As the penetration block becomes significant after the fusion block is established, only the latter effectively prevents polyspermy in mice. We propose that it acts through neutralization of excess sperm in the PVS by oocyte-derived proteins CD9 and JUNO coating non-fertilizing spermatozoa.

**Keywords** Gamete Fusion; Mammalian Fertilization; Polyspermy Prevention; Real Time Imaging; Sperm Flagellum Beating
**Subject Category** Development

## Introduction

In mammals, fertilization is the process by which female and male gametes interact and fuse to form a single cell called a zygote (Fig. 1A,B). Once a spermatozoon has passed through the cumulus cell layer surrounding the oocyte, it still must overcome two oocyte's barriers to complete fertilization. The first is the zona pellucida (ZP), the outer glycoprotein envelope of the oocyte that the spermatozoon must traverse to penetrate into the perivitelline space (PVS) and reach the second barrier: the oolemma. The oolemma is the plasma membrane of the oocyte with which the spermatozoon head must interact and fuse to achieve fertilization. To overcome these two barriers, the oocyte and the spermatozoon must exhibit complementary functional properties. This cooperation is reflected in the interplay between the spermatozoon's motility and the adhesive properties of both gametes. The beating of the flagellum enables the spermatozoon to reach the oocyte, to traverse the ZP and position its head optimally for fusion with the oolemma. At the same time, the ability of the spermatozoon head and the oocyte surfaces—first the ZP, then the oolemma—to adhere together ensures that the flagellar movement drives the spermatozoon through the ZP and maintains the contact between the spermatozoon head and the oolemma, rather than causing the spermatozoon to drift away from the oocyte and then to detach from the oolemma (Baltz et al, 1988; Bleil et al, 1988; Bleil and Wassarman, 1980). Moreover, previous studies conducted in mice suggest that the usefulness of flagellum beating is not limited to its ability to bring the spermatozoon to the site of fertilization, but also directly contributes to making the interaction between the spermatozoon head and the oolemma conducive to fusion (Ravaux et al, 2016, 2018). Indeed, these studies revealed a striking correlation between vigorous flagellum beating - which, unlike weaker or erratic beating or mechanical inhibition of vigorous beating, produced push-up-like movements of the spermatozoon head against the oolemma- and molecular membrane remodeling at the gamete contact zone, conducive to fusion (Ravaux et al, 2018). Notably, this remodeling involved CD9 and JUNO, the two oolemma proteins currently identified as crucial for gamete fusion, as well as IZUMO1, the receptor for JUNO on the spermatozoon head, which is also essential for fusion (Bianchi et al, 2014; Kaji et al, 2000; Le Naour et al, 2000; Miyado et al, 2000; Ravaux et al, 2018). However, as these previous studies were performed on ZP-free oocytes, flagellar movement was unrestricted. In contrast, the physiological presence of the ZP imposes physical constraints on flagellum dynamics and, consequently, on the movement of the spermatozoon head on the oolemma during interaction. This

[1]Laboratoire de Physique de l'Ecole Normale Supérieure, ENS, Université PSL, CNRS, Sorbonne Université, Université Paris Cité, F-75005 Paris, France. [2]Université Paris Cité, Institut Cochin, INSERM, CNRS, F-75014 Paris, France. [3]Département de biologie, Ecole Normale Supérieure, Institut de Biologie de l'ENS (IBENS), CNRS, INSERM, Paris, France. [4]Service d'histologie, d'embryologie, Biologie de la Reproduction, AP-HP, Hôpital Cochin, F-75014 Paris, France. [5]These authors contributed equally: Sophie Favier, Nathan Martin-Fornier. ✉E-mail: nicolas.rodriguez@ens.fr; christine.gourier@ens.fr

restriction may arise either because the flagellum remains partially embedded in the ZP at this stage, or because it is confined to the narrow PVS. The extent to which the previously inferred promoting-fusion role of flagellum beating is modulated under physiological conditions provided by ZP-intact oocytes, therefore, remains a key open question. It also raises the intriguing possibility of a direct involvement of the ZP in the fusion event itself, a hypothesis that has been investigated in this study.

In mammals, oocytes fertilized by more than one spermatozoon cannot develop into viable offspring. If two or more spermatozoa manage to reach the PVS of an oocyte and more than one of them fuses with the oolemma, polyspermy occurs. To prevent such a detrimental situation, firewalls triggered by the first fertilization are deployed by the new zygote to stop the progression of any additional spermatozoa on their way to fertilization at one of the oocyte's two barriers. These firewalls are the fertilization-induced responses of the oocyte, leading to functional modifications in its three structural layers: the ZP, the PVS and the oolemma (Fig. 1C) (Talbot and Dandekar, 2003; Evans, 2020; Fahrenkamp et al, 2020). Alterations of the ZP that impede spermatozoa from traversing it are referred to as ZP-block (Fahrenkamp et al, 2020) and contribute to a penetration block. Similarly, modifications in the PVS or oolemma that diminish the probability of spermatozoa reaching the PVS to successfully fuse with the oolemma are referred to as PVS-block (Talbot and Dandekar, 2003) and membrane-block (Evans, 2020) respectively and both contribute to a fusion block (Fig. 1C). The variability across mammalian species in both the rate of fertilized oocytes with additional spermatozoa in their PVS (from 0 to more than 80%) after natural mating and the number of spermatozoa present in the PVS of these oocytes (from 0 to more than a hundred) (Braden and Austin, 1954) suggests that the time for completion of the penetration block and thus its efficiency to prevent polyspermy can vary significantly between species. Surprisingly, while each additional spermatozoon in the PVS inherently poses a risk of polyspermy, polyspermy remains very low across species (Braden and Austin, 1954). This suggests that, when required, the fusion block can efficiently compensate for the relative weakness of the penetration block in preventing polyspermy.

Viable monospermic fertilization can therefore be conceptualized as a two-phase process involving a single oocyte and multiple spermatozoa. In phase 1, the oocyte and spermatozoa cooperate by exhibiting complementary properties that enable the occurrence of a first fertilization event. In phase 2, this cooperation shifts to a competitive dynamic, wherein the new zygote must outpace all remaining spermatozoa that could jeopardize its monospermic status. It does so by actively modifying the properties of its structural components, resulting in the establishment of both a penetration block and a fusion block. The respective contributions of these two blocks to the prevention of polyspermy and their ability to maintain the monospermic status of the zygote ultimately depend on their relative dynamics—that is, how rapidly and effectively each block is established in response to fertilization.

This study aims to quantify the kinetics of fertilization and subsequent penetration and fusion blocks. It also aims to elucidate the actual roles of each component of the spermatozoon (flagellum and head) and oocyte (ZP, PVS, and oolemma) in both enabling fertilization and preventing polyspermy. Real-time imaging of individual ZP-intact inseminated oocytes has proven to allow the

tracking of spermatozoa as they traverse the ZP, penetrate the PVS, and fuse with the oolemma (Sato, 1979; Jin et al, 2011). Although not often used to study the kinetics of fertilization and the establishment of polyspermy block, this approach remains the only one capable of capturing transient events within these processes and characterizing their dynamic nature under conditions as close to physiological as possible. To this end, this study applies real-time brightfield imaging completed by confocal imaging to dozens of individually inseminated ZP-intact mouse oocytes. The visual information we obtained from more than a hundred spermatozoa, correlated with a statistical analysis of the time of their penetration into the PVS and, when relevant, the time of their fusion, allow us to establish a precise dynamic picture of the sequence of events leading to fusion and prevention of polyspermy, and of the role of each gamete component in these processes. The kinetics we have determined challenge prevailing assumptions about the actual role of the ZP in blocking polyspermy in mice. They suggest a key role for a previously underappreciated PVS-block in maintaining monospermy. They also reveal a mechanical function of the ZP in channeling gamete interaction that leads to fusion. These discoveries made in mice open up new avenues for research into fertilization and the prevention of polyspermy in mammals.

## Results

Throughout this study, we will discuss two different events involved in fertilization that are often referred to indifferently in literature as "penetration". The first corresponds to the entry of a spermatozoon into the perivitelline space (PVS) of an oocyte, following its traversal of the zona pellucida (ZP). The second event refers to the fusion of the spermatozoon head with the oolemma, followed by its internalization into the ooplasm. To avoid ambiguity, we have chosen to use the term "penetration" exclusively for the first event and "fertilization" or "fusion" for the second. Accordingly, a spermatozoon crossing the ZP and entering the PVS is referred to as a "penetrating spermatozoon" once its head is visible in the PVS (Fig. 1B). If it subsequently fuses with the oolemma, it is additionally classified as a "fertilizing or fusing spermatozoon". This experimental study was conducted in mice, which are the most widely used model for studying fertilization and the prevention of polyspermy in mammals. While there are many interspecies similarities, the findings presented here should not be directly extrapolated to humans or other mammalian species without species-specific validation.

### Comparative study of penetration and fertilization rates under in vivo and in two distinct in vitro fertilization conditions

The quantity of spermatozoa reaching an oocyte and the timing at which they make contact with it strongly depend on the insemination conditions. These quantitative and kinetic factors are crucial, as they are likely to influence the number of spermatozoa penetrating the PVS of an oocyte, the number of fertilizations and, consequently, the ability of a zygote to develop into a viable pup. We conducted a comparative study on the number of penetrations in the PVS and the number of subsequent

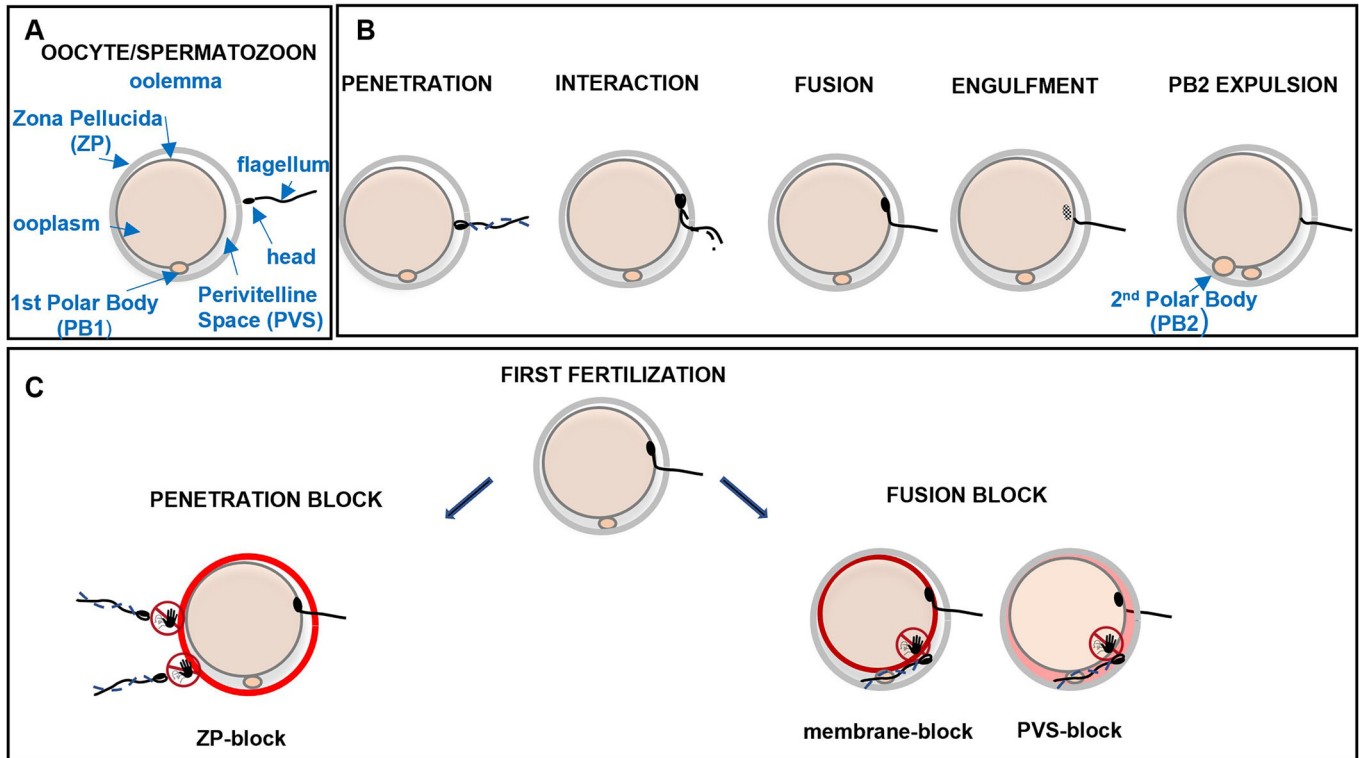

**Figure 1.  Fertilization process and fertilization-induced responses of the oocyte contributing to the penetration and fusion blocks.**

(**A**) Oocyte and spermatozoon components. (**B**) Sequence of events involved in successful fertilization: penetration of the spermatozoon in the PVS, interaction of the spermatozoon head with the oolemma, fusion of the spermatozoon head with the oolemma, engulfment of the spermatozoon head in the ooplasm, expulsion of the second polar body (PB2). (**C**) Fertilization-induced responses of the oocyte. In different shades of red: modifications in the ZP properties that impede spermatozoa from traversing it are referred to as ZP-block and contribute to a penetration block, and modifications in the oolemma and PVS properties that diminish the probability of spermatozoa reaching the PVS to successfully fuse with the oolemma are referred to as PVS-block and membrane-block, respectively, and both contribute to the fusion block.

fertilizations in more than 500 oocytes inseminated in three different conditions (Fig. 2A). Condition 1 corresponds to physiological in vivo inseminations resulting from mating: 211 oocytes were recovered from superovulated females 15 h after mating. Condition 2 corresponds to standard in vitro fertilization (220 oocytes): in each experiment, the cumulus–oocyte complexes of two superovulated mice were inseminated for 4 h with capacitated spermatozoa at a concentration of $10^6$ /mL. Condition 3 corresponds to in vitro fertilizations with kinetics tracking (93 oocytes): after 15 min of incubation of cumulus–oocyte complexes under the previous standard in vitro fertilization conditions, some oocytes were collected with the spermatozoa already bound to their ZP to individually track their evolution -sperm penetrations in the PVS, flagellum beating patterns, movement of the sperm head on the oolemma, adhesion to the oolemma, fusion, sperm engulfment, PB2 release in real-time for 4 h.

## The birth of viable offspring from monospermic zygotes with additional spermatozoa in the PVS demonstrates the physiological significance of the fusion block

Condition 1, corresponding to in vivo fertilization experiments, provides reference values for the frequency of multi-penetration and polyspermy events under physiological conditions, where

ejaculated spermatozoa gradually reach ovulated oocytes over time (Jin et al, 2011; Austin and Braden, 1952; Hunter, 1996). These experiments reveal that, despite natural regulation, multi-penetration of spermatozoa in the PVS is quite common among fertilized oocytes: $16 \pm 6\%$ (Fig. 2B) are penetrated by two to three spermatozoa (Fig. 2D). However, all of these oocytes remained monospermic (Fig. 2C). This suggests that the fusion block (Fig. 1C) efficiently prevents the fusion of additional spermatozoa present in the PVS. However, the contribution of the fusion block to prevent polyspermy has physiological significance only if monospermic oocytes with additional spermatozoa in their PVS can develop into viable pups. To investigate this, we implanted (i) 16 two-cell stage monospermic zygotes, containing additional non-fused spermatozoa in their PVS, into one pseudo-pregnant female (a female mated with a sterile male to prepare the female for pregnancy) and (ii) 16 two-cell stage monospermic zygotes per female, with no extra sperm in the PVS, into two other pseudo-pregnant females. The lack of differences in litter sizes (6, 6, and 7 pups respectively) confirms that the presence of spermatozoa in the PVS of monospermic zygotes does not hinder the birth of viable pups. This underscores the physiological significance of the fusion block in preventing polyspermy in oocytes penetrated by multiple spermatozoa and justifies the usefulness of the characterization of the fusion block.

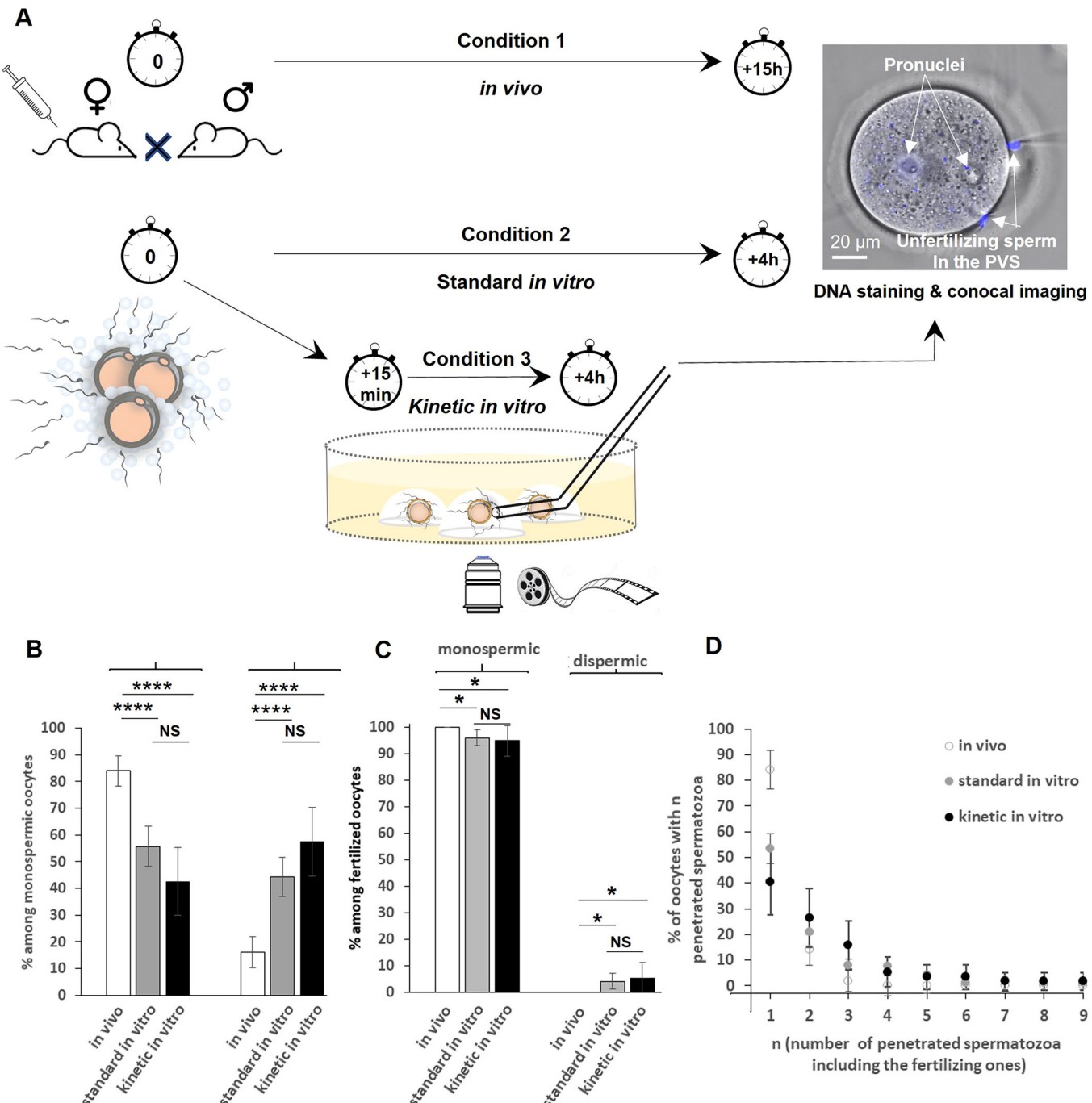

**Insufficient penetration block is complemented with good efficiency by the fusion block to prevent polyspermy**

Condition 2, corresponding to standard in vitro fertilization (Fig. 2A) differs from the previous in vivo fertilization Condition 1 in that oocytes are quickly surrounded by a large number of spermatozoa, which simultaneously bind to their ZP and can be replaced by others if they detach. In this sperm-rich environment, the number of sperm penetrations into the PVS of oocytes is not limited by the availability of spermatozoa for penetration, as it

might be in vivo. Condition 2 guarantees that the maximum possible number of spermatozoa will penetrate the PVS. Thus, it represents the ideal condition to assess the effectiveness of both the penetration block and the fusion block. In this sperm-rich environment, we observed that multi-sperm penetrations in the PVS occur in $47 \pm 8\%$ of fertilized oocytes (Fig. 2B "standard in vitro"), with the number of spermatozoa penetrating these oocytes ranging from 2 to 9 (Fig. 2D). Given the large number of spermatozoa potentially available to traverse the ZP, these figures remain relatively low, highlighting the critical role of the ZP in limiting the number of sperm entering the PVS. However, since any

◄ **Figure 2. Multi-penetration of spermatozoa in the PVS of oocytes and polyspermy as a function of the experimental insemination conditions.**

(A) Experimental conditions. Condition 1 ("in vivo" $N = 211$ oocytes): mouse females are superovulated, mated, and oocytes are recovered 15 h after mating and stained with Hoechst. Condition 2 ("standard in vitro" $N = 220$ oocytes): cumulus–oocyte complexes of two superovulated are inseminated in vitro with capacitated spermatozoa ($10^6$/mL). Four hours later, the oocytes are washed and stained with Hoechst. Condition 3 ("kinetic in vitro" $N = 93$): after 15 min of incubation in Condition 2, some oocytes and bound spermatozoa are isolated in individual drops, and imaged in turn for 4 h in bright field to collect temporal and visual information about penetration of spermatozoa in the PVS, interaction with the oolemma, fusion, sperm internalization, PB2 release. The oocytes are then stained with Hoechst. For all three conditions, the oocytes are imaged to determine the number of penetrations and fusions in each oocyte. Scale bar 20 μm. (B) Comparison between the three experimental insemination conditions of the rate of monospermic oocytes ($N = 156$ in vivo, $N = 165$ standard in vitro, $N = 54$ kinetic in vitro) without extra spermatozoa in the PVS or with extra sperm in the PVS. (C) Comparison between the 3 experimental insemination conditions of the rate of fertilized oocytes ($N = 156$ in vivo, $N = 172$ standard in vitro, $N = 57$ kinetic in vitro), which are monospermic (1 fertilization) and dispermic (2 fertilizations). (D) Comparison between the three experimental insemination conditions of the rate of fertilized oocytes ($N = 156$ in vivo, $N = 172$ standard in vitro, $N = 57$ kinetic in vitro) with n penetrated spermatozoa, including the fertilizing one(s). In (B–D), the error bars correspond to 95% confidence intervals (±1.96 SEM) and. In (B, C), p values correspond to Chi-squared tests. In (B) standard in vitro/kinetic in vitro NS $p = 0.127$, in vivo/standard in vitro and in vivo/kinetic in vitro ****$p < 0.0001$. In (C) standard in vitro/kinetic in vitro NS $p = 0.99$; in vivo/standard in vitro *$p = 0.03$; in vivo/kinetic in vitro *$p = 0.026$. Source data are available online for this figure.

spermatozoon that penetrates the PVS of an oocyte beyond the first may produce polyspermy, the penetration of two to nine spermatozoa in nearly half of fertilized oocytes represents a significant potential for polyspermy. However, we found that $96 \pm 3\%$ of fertilized oocytes remain monospermic, and the few cases of polyspermy are limited to dispermy (Fig. 2C). This shows that although multi-penetration is frequent and the index of penetration is high, the fusion block prevents polyspermy with good efficiency.

## The number of spermatozoa penetrating the PVS and polyspermy depend more on the number of spermatozoa initially bound to the ZP of an oocyte than on the total number of spermatozoa reaching the oocyte over time

Condition 3, corresponding to in vitro fertilization with kinetics tracking (Fig. 2A), differs from standard in vitro fertilization Condition 2 in that the number of spermatozoa likely to traverse the ZP and enter the PVS of an oocyte is limited to those already bound to the ZP (typically a few dozen) when the oocytes were isolated 15 min after insemination. In terms of the total number of spermatozoa that come into contact with the oocyte over time, Condition 3 is likely closer to physiological in vivo fertilization Condition 1 than to standard in vitro fertilization Condition 2, where the reservoir of spermatozoa likely to enter into contact with the oocyte is almost unlimited. However, in terms of the number of spermatozoa quickly and simultaneously bound to the ZP of an oocyte, Condition 3 is identical to Condition 2 and different from in vivo fertilization Condition 1, where spermatozoa reach the oocytes more gradually over time. We have observed that (i) the rates of mono- and multi-penetrated fertilized oocytes (Fig. 2B), (ii) the distribution of the number of spermatozoa penetrated in the PVS of these oocytes (Fig. 2D), and (iii) the rates of mono- and dispermic oocytes (Fig. 2C) are similar for standard and kinetic in vitro fertilizations but significantly different from in vivo fertilizations. We can therefore deduce that multi-penetration in the PVS and polyspermy depend more on the number of spermatozoa rapidly and simultaneously bound to an oocyte than on the total number of spermatozoa reaching an oocyte over time. Furthermore, the lack of significant differences in the outcomes of Conditions 2 and 3 (Fig. 2B–D) indicates that in vitro fertilization with kinetics tracking is as suitable as standard in vitro fertilization for assessing the effectiveness of both penetration and fusion

blocks. In addition, unlike in vivo and standard in vitro fertilizations, which are performed blindly, in vitro fertilization with kinetic tracking can provide valuable temporal and visual insights into key transient events related to fertilization and block to polyspermy. Therefore, we will employ this approach hereinafter to characterize the kinetics of fertilization, as well as the kinetics of the penetration and fusion blocks.

## In vitro fertilization with kinetic tracking reveals the timing of penetration and fusion events in each oocyte

Out of the 93 oocytes subjected to in vitro fertilization with kinetic tracking (Condition 3), 57 were penetrated by a total of 138 spermatozoa and fertilized by 60 of them, 14 were penetrated by 46 spermatozoa but not fertilized, and 22 were not penetrated at all. For each spermatozoon penetrating into the PVS of an oocyte, we could experimentally determine a penetration time window during which the spermatozoon penetrated into the PVS of its oocyte. For spermatozoa that also fertilized, we could additionally determine a fusion time window during which gamete fusion occurred. These penetration and fertilization time windows are illustrated by solid and dashed segments, respectively reported in two chronograms: one for fertilized oocytes (Fig. 3A) and one for the oocytes that remained unfertilized despite one or more spermatozoa have penetrated their PVS (Appendix Fig. S1A). A color code allows for easy identification of each spermatozoon, based on whether it was fertilizing (black) or a non-fertilizing spermatozoon. Non-fertilizing spermatozoa are further categorized as having penetrated the PVS before the first fertilizing spermatozoon (green) or after fertilization (purple). Yellow is used for the spermatozoa for which it was not possible to distinguish between the two previous scenarios.

## Being the first penetrating spermatozoon is not sufficient to be a fertilizing spermatozoon

Among the 57 spermatozoa that were the first to penetrate the PVS of their oocyte, 63.2% successfully fertilized the oocyte, 14% failed to do so, and for the remaining 22.8, it remains uncertain whether they were among the spermatozoa that fertilized their oocytes (Fig. 3B). These findings indicate that being the first spermatozoon to enter the PVS of an oocyte is not sufficient to be the spermatozoon that will ultimately fertilize this oocyte.

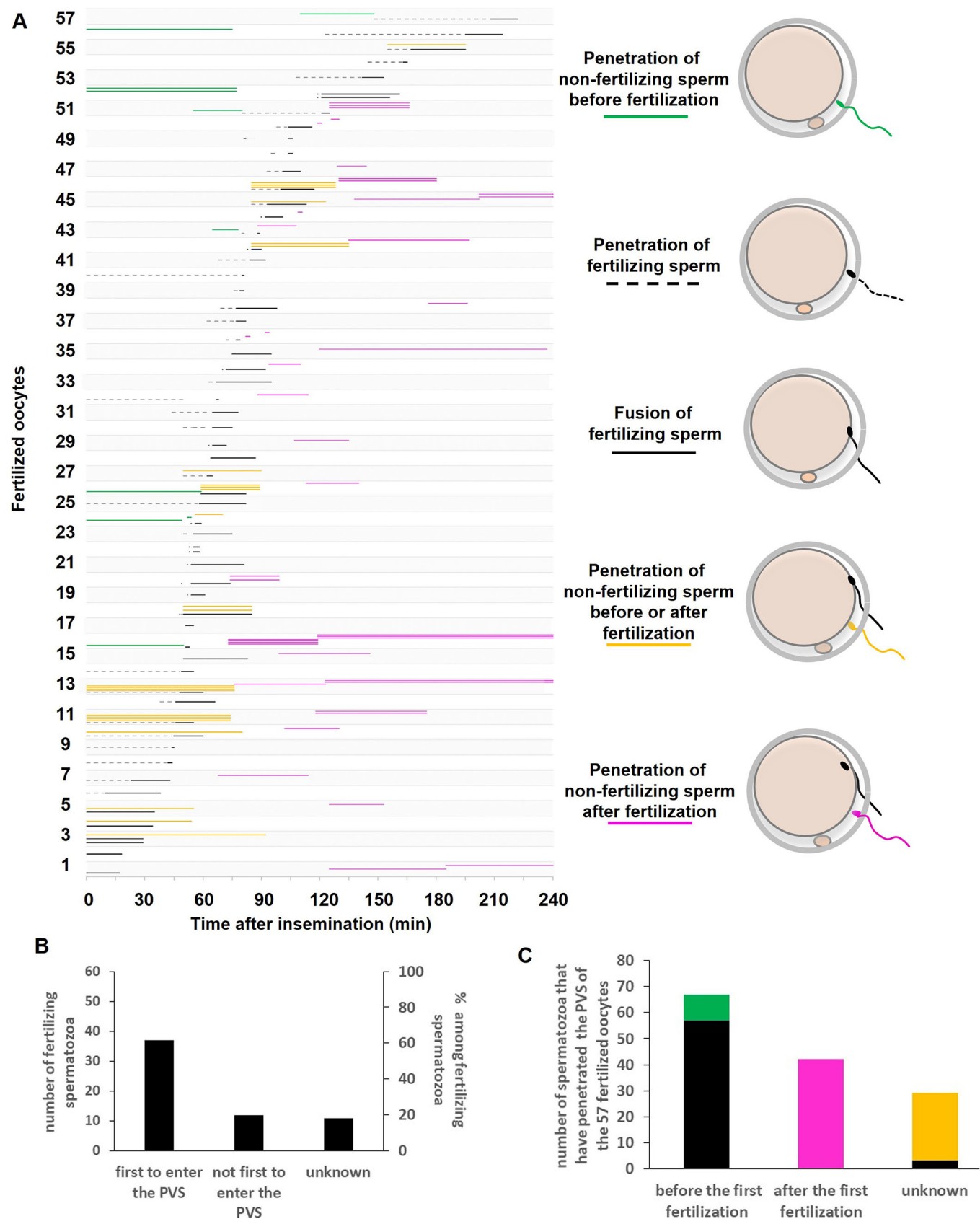

**Figure 3.  Sequence of penetration and fertilization events.**

(A) Chronogram of penetration and fertilization events in fertilized oocytes (57 oocytes, 138 penetrated spermatozoa, including 60 fertilizing spermatozoa). Each strip corresponds to one oocyte. Each line within a strip corresponds to a different spermatozoon that has penetrated the PVS of this oocyte. For each penetrating spermatozoon, we could experimentally determine a penetration time window during which the spermatozoon penetrated the PVS of its oocyte. For those that fertilized the oocyte, we could also determine a fusion time window during which gamete fusion occurred. The penetration and fertilization time windows of a fertilizing spermatozoon are represented by dashed black and solid black lines, respectively. The penetration time windows of the non-fertilizing spermatozoa are represented by colored lines: green for those that penetrated the PVS of their oocyte even before the first fertilizing spermatozoon, purple for those that entered the PVS after the oocyte has already been fertilized, and yellow for spermatozoa whose penetration time window overlaps both the penetration and fusion time windows of the first fertilizing spermatozoon. In this latter case, it was not possible to determine which spermatozoon (fertilizing or non-fertilizing) entered the PVS first, or whether the non-fertilizing spermatozoon entered the PVS before or after the fertilizing spermatozoon has fused with the oolemma. (B) The rate of the first penetrated spermatozoa that successfully fertilized. (C) Distribution of spermatozoa that have penetrated before and after fertilization, and among them, the number of fertilizing spermatozoa. The black bars represent the fertilizing spermatozoa; the green bar represent the unfertilizing spermatozoa that penetrated their oocyte before it was fertilized by another spermatozoon; the purple bars correspond to the unfertilizing spermatozoa that have entered the PVS after their oocyte has been fertilized; the yellow bars correspond to the unfertilizing spermatozoa for which it was not possible to determine whether they entered the PVS of their oocyte before or after the latter has been fertilized. Source data are available online for this figure.

## The first fertilization is highly effective at blocking subsequent fertilizations but less so at preventing new penetrations

Out of the 138 spermatozoa that entered the PVS of the 57 fertilized oocytes, 67 did so while the oocyte was still unfertilized, among which 85.1% (57/67) successfully fused. In contrast, none of the 42 spermatozoa penetrated after the first fertilization fused (0/42). For the remaining 29 spermatozoa, the timing of their entry into the PVS relative to fertilization of the oocyte is unknown. Among them, 10.3% (3/29) fused as a second fertilizing spermatozoon, leading to the formation of 3 dispermic zygotes (Fig. 3C). These results show that a spermatozoon that penetrates a yet unfertilized oocyte has high chance of fertilizing it, but its chance dramatically decreases if another spermatozoon (whether penetrated before or after it) manages to fertilize the oocyte first. As for the many spermatozoa penetrating after the first fertilization event, their chance to fertilize is almost zero. This suggests that the first fertilization is much more efficient at blocking subsequent fertilizations (whether by spermatozoa already present in the PVS or by those that will enter later) than at preventing additional sperm penetrations in the PVS.

## The choreography of the fertilizing spermatozoa: from penetration in the PVS to fusion with the oolemma

In addition to providing the timing of the penetration and fertilization events, in vitro fertilization with kinetic tracking allows for the observation of the choreography of the spermatozoa as they enter the PVS of their oocyte and interact with the oolemma. The typical choreographies of a spermatozoon penetrating the PVS of an unfertilized oocyte and fertilizing it are provided by Movies EV1 and EV2, with the main stages from penetration in the PVS to fusion, illustrated in Fig. 4A. As the spermatozoon passes through the ZP and starts to penetrate into the PVS, the tip of its head quickly bumps into the oolemma. The spermatozoon may remain stuck in this position perpendicular to the oolemma for several minutes (~6 min in Movie EV1), with its head oscillating in a wiping motion, the tip grazing the surface of the oolemma, before the persistent flagellum movements allow the spermatozoon to move further into the PVS. This causes the spermatozoon head to orient in a more parallel position relative to the oolemma and to start adhering to the oolemma. As long as this adhesion remains

weak, the spermatozoon may keep on progressing along the oolemma by small hops. This "hopping" phase can last several minutes (~2 min in Movie EV1) before the increasing adhesion and the movement of the flagellum, still trapped in the ZP, become strong enough to transform the hops into "push-ups" of the sperm head against the oolemma. This phase typically lasts for a few minutes (~1.5 min in Movie EV1) and culminates in the fusion of the fertilizing spermatozoon, identified by the stop of head movements on the oolemma due to arrest of flagellum oscillations, and confirmed in the following minutes by the observation of the progressive engulfment of the sperm head into the ooplasm (Appendix SI2; Appendix Fig. S2A,C), followed by the expulsion of the second polar body (Appendix Fig. S2B,C) marking the end of meiosis. A majority of fertilizing spermatozoa (73%) fused with the oolemma while their flagellum was still partially embedded in the PVS, as shown in Movie EV1. The remaining 27% completed fusion after their flagellum has fully entered the PVS, as shown in Movie EV2. Notably, in both situations, the spermatozoon head exhibited characteristic push-up movements against the oolemma in the minutes leading up to fusion (Movies EV1 and EV2), in contrast to spermatozoa fully entered in the PVS that failed to fertilize (Movies EV3 and EV4).

## Flagellum entrapment in ZP promotes sperm head movement on the oolemma, leading to fusion

The choreographies of all spermatozoa penetrating the PVS of still unfertilized oocytes appear similar as long as their flagellum remains partially embedded in the ZP. On one hand, the entrapment of the flagellum within the ZP, and on the other, the binding of the spermatozoon head to the oolemma, together restrict the motion of the flagellum segment located in the PVS and, consequently, the movements of the sperm head on the oolemma. Once the flagellum fully extricates itself from the ZP, the constraint imposed by its previous entrapment is lifted (Fig. 4B). The flagellum can then move more freely, restricted only by the boundaries of the PVS (the oolemma and the ZP) and the binding of its head to the oolemma. Compared to earlier entrapment in the ZP, these boundaries impose significantly fewer restrictions on flagellum movement, as demonstrated by the vigorous (Movies EV2 and EV3) or erratic (Movie EV4) movements exhibited by the flagellum of these spermatozoa. These movements are transmitted to the spermatozoon head and influence the interaction with the

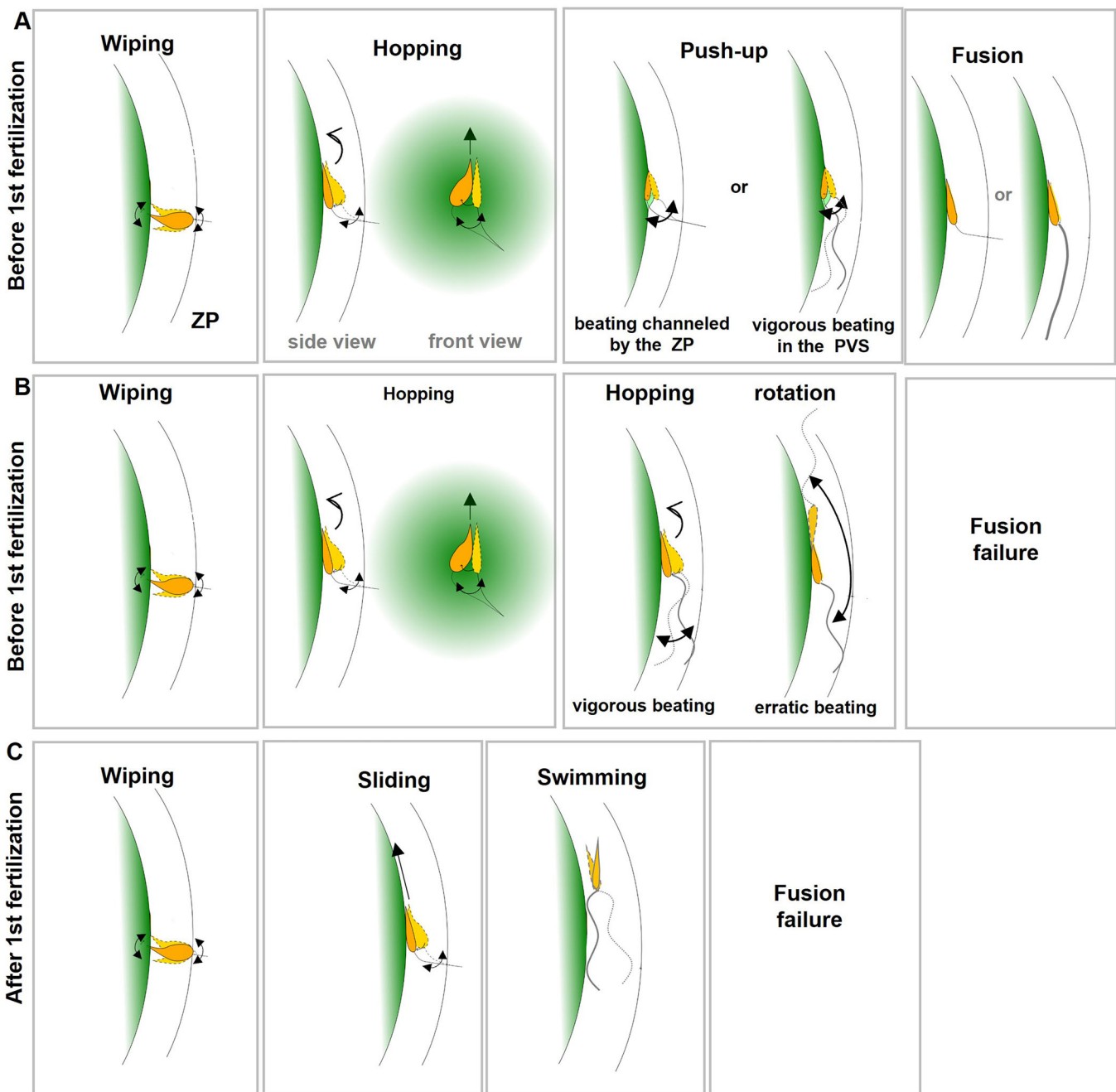

**Figure 4. The choreographies of the fertilizing and non-fertilizing spermatozoa.**

(A) Typical three-phase choreography of a fertilizing spermatozoon entering the PVS of an unfertilized oocyte: (Wiping phase) the spermatozoon head is oriented perpendicular to the oolemma and oscillates in a wiping motion; (Hopping phase) once parallel to the oolemma, the spermatozoon head starts to interact with the oolemma and progresses in small hops over it; (Push-up phase left) the flagellum oscillations within the PVS, restricted by the adhesion of the sperm head on one side and the entrapment of the flagellum on the other, forces the head into a push-up-like motion, ultimately leading to fusion; (Push-up phase right) alternately, sperm flagellum may have fully traversed the ZP during the hopping phase. Freed from its entrapment in the ZP, it oscillates more freely. Fusion can occur only when flagellum oscillations display vigorous movements compatible with a push-up-like motion of the spermatozoon head against the oolemma. (B) Typical choreography of a non-fertilizing spermatozoon entering the PVS of an unfertilized oocyte: following the wiping phase, the sperm flagellum fully enters the PVS during the hopping phase. The flagellum may exhibit vigorous or erratic oscillations, causing the sperm head to move or pivot on the oolemma in a manner incompatible with membrane fusion. (C) Typical choreography of a non-fertilizing spermatozoon entering the PVS of a fertilized oocyte: following the wiping phase, the sperm head aligns parallel to the oolemma but slides over it without adhering. Once fully within the PVS, the spermatozoon swims around the oolemma and does not fuse.

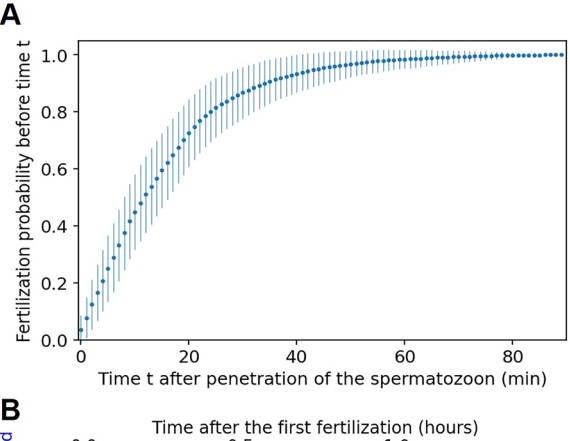

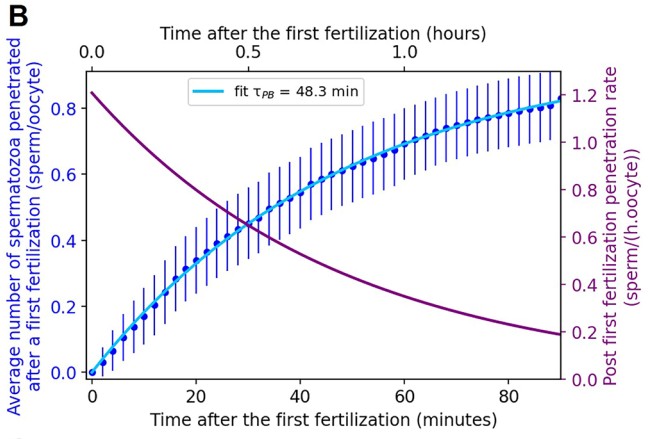

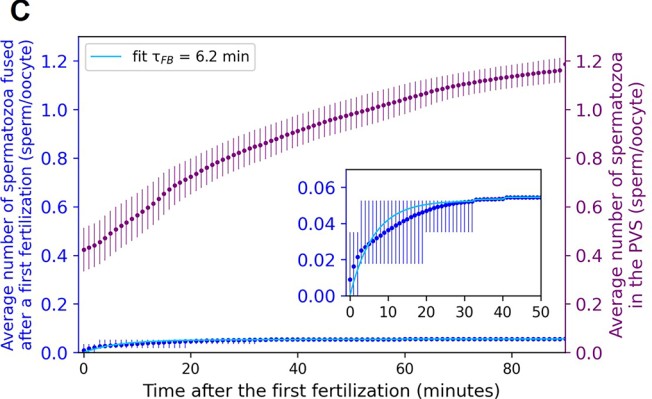

Figure 5. Kinetics of fertilization, penetration block, and fusion block.

(A) Probability for a fertilizing spermatozoon to have fused with the oolemma before time t after its penetration in the PVS of an oocyte (blue dots). (B) Average number per fertilized oocyte of spermatozoa penetrating an oocyte after it has been fertilized (dark blue dots). The fitting curve (light blue line) shows that after the first fertilization, the ZP becomes impermeable to spermatozoa with a time constant $\tau_{PB}$ = 48.3 ± 9.7 min. The rate of sperm penetrating the PVS after a first fertilization (solid purple line), obtained by derivation of the blue fitting curve, reflects the mean time evolution of ZP to spermatozoa after a first fertilization. Its initial value (1.21 ± 0.37 sperm/oocyte/h at time zero) shows the low ZP permeability baseline before any fertilization-related processes contribute to the penetration block. (C) Average number of spermatozoa per fertilized oocyte that fuse with an already fertilized oocyte (dark blue dots). Insert is a zoom of the first 50 min. The corresponding fitting curve (light blue line) shows that after a first fertilization, the fusion block takes place with a time constant $\tau_{FB}$ = 6.2 ± 1.3 min. The average number of non-fused spermatozoa present in the PVS of fertilized oocytes vs time after a first fertilization is also shown (purple dots). In (A–C), all the curves were obtained by statistical analysis of fertilized oocytes' penetration and fertilization chronogram (Fig. 3A), and the error bars correspond to a 95% confidence interval. Source data are available online for this figure.

promoting gamete fusion by channeling flagellum oscillations. This also shows that, regardless of any fertilization-triggered fusion block, certain spermatozoa are doomed to fail at fusion due to inappropriate flagellum beating if they do not succeed to fuse before fully entering the PVS.

## Impaired adhesion between penetrating spermatozoa and the fertilized oocyte oolemma

Unlike the spermatozoa penetrating into the PVS of unfertilized oocytes whose head consistently adheres to the oolemma (Movies EV1–EV4), the spermatozoa penetrating the PVS of fertilized oocytes do not firmly attach to the oolemma and instead tend to slide over it (Movie EV5; Fig. 4C). As illustrated in Movie EV5 which captures the entry of a spermatozoon into the PVS of a fertilized oocyte, the adhesion of the penetrating spermatozoon during the hopping phase is so weak that it does not hinder the spermatozoon rapid progression through the PVS. Once its flagellum is fully released from the ZP, the spermatozoon glides along the oolemma and begins to swim in the PVS. Interestingly, most of the non-fertilizing spermatozoa found in the PVS of fertilized oocytes are observed swimming freely around the oolemma, with only a few seen attached to the oolemma. Deficient adhesion to the oolemma thus appears to be the main cause of the fertilization failure for spermatozoa that penetrate the PVS of fertilized oocytes and therefore the main contribution to the fusion block.

## Statistical treatment of penetration and fertilization chronograms to study the kinetics of fertilization, penetration block and fusion block

After a first fertilization, the preservation of monospermy depends on the kinetics of fusion as compared with the kinetics of the penetration and/or fusion blocks. To investigate this, we analyzed our large dataset comprising 57 fertilized oocytes penetrated by 138 spermatozoa, and 14 unfertilized oocytes penetrated by

oolemma in ways that can either promote (Movie EV2) or hinder (Movie EV4) gamete fusion. Among the spermatozoa that have fully entered the PVS, only those exhibiting vigorous flagellum oscillations and robust adhesion of the head to the oolemma were able to perform the push-up-like head movements that lead to fusion (Movie EV2). Those displaying vigorous oscillations but weak head adhesion failed to fuse (Movie EV3; Fig. 4B), as did spermatozoa displaying erratic beating patterns that caused abrupt head rotations tangentially to the oolemma (Movie EV4; Fig. 4B). These observations show that successful fusion requires flagellum oscillations compatible with a push-up-like motion of the head against the oolemma. While ZP entrapment channels these oscillations in a way compatible with such movements, this favorable constraint is lost once the flagellum fully enters the PVS. These observations highlight an unsuspected role of the ZP in

46 spermatozoa. Using the experimentally determined penetration and fusion time windows (Fig. 3A; Appendix Fig. S1A), we were able to estimate both the average kinetics of fertilization and the dynamics governing the establishment of the penetration and fusion blocks.

Since the actual time of penetration or fusion within each time window could not be determined, we assumed a uniform probability distribution across the entire interval. In other words, we had no reason to consider one time point within the window more likely than another. Based on this assumption, we computed average penetration and fusion times by sampling all equiprobable points within each time window for each spermatozoon. For each oocyte, we then calculated the mean kinetics of penetration and fusion from its individual sperm data. Finally, these oocyte-level values were averaged across all fertilized and unfertilized oocytes, yielding overall estimates of penetration and fusion kinetics, as well as their associated uncertainties, in both populations.

## On average, a fertilizing spermatozoon spends around 15 min in the PVS before fusing

To study the kinetics of fertilization, from penetration of the fertilizing spermatozoon to its fusion with the oolemma, we determined the probability that the fertilizing spermatozoon would fuse before time $t$ after its penetration into the PVS of an oocyte (Fig. 5A). This termination reveals that on average, the time between the penetration of a fertilizing spermatozoon into the PVS of an oocyte and its fusion with the oolemma is $15.8 \pm 5.7$ min. The relative slowness and variability of the time required for a spermatozoon to achieve fusion after penetration provide an explanation of why a spermatozoon penetrated in the PVS after the first one can fertilize the oocyte first.

## Sperm entry into the PVS is regulated by a ZP low baseline permeability before fertilization and by the slow kinetics of the penetration block thereafter

To study the kinetics of the penetration block, we quantified the average number of spermatozoa that entered the PVS of an oocyte over time following the first fertilization event (dark blue dots in Fig. 5B). This number was found to increase at a progressively slower rate, indicative of a gradual decline in the permeability of the ZP to spermatozoa as time progresses. The temporal evolution of this process can be accurately described by the following function:

$$f(\mathrm{t}) = A\left(1 - \exp\left(\frac{-\mathrm{t}}{\tau_{PB}}\right)\right) \qquad (1)$$

where $A$ represents the average number of spermatozoa that penetrate the PVS of a fertilized oocyte before its ZP becomes fully impermeable, and $\tau_{PB}$ is the time constant of the penetration block. The best fit (light blue line in Fig. 5B) is obtained for $A = 0.97 \pm 0.10$ spermatozoa/oocyte and $\tau_{PB} = 48.3 \pm 9.7$ min. The corresponding time-dependent sperm penetration rate into the PVS, derived from this fit (purple line in Fig. 5B), directly illustrates the progressive decline in the ZP permeability to spermatozoa due to the penetration block. In contrast, the constant sperm penetration rate obtained in unfertilized oocytes (Appendix Fig. S1B) demonstrates that sperm

penetration alone does not alter ZP permeability and confirms that fusion alone is the trigger for the onset of the penetration block. Moreover, the similar values obtained for the steady penetration rate in unfertilized oocytes ($1.28 \pm 0.07$ sperm/oocyte/h) (Appendix Fig. S1B) and the initial penetration rate at the time of the first fertilization in fertilized oocytes ($1.21 \pm 0.37$ sperm/oocyte/h at $t = 0$) (Fig. 5B) confirm that the ZP of unfertilized oocytes displays normal permeability. These two independent determinations define the baseline permeability of the ZP prior to any fertilization-induced modifications. Its intrinsically low value makes the mouse ZP an effective barrier to sperm penetration even prior to fertilization, thereby reducing the likelihood of simultaneous entries into the PVS and therefore the risk of polyspermy.

## The fusion block is established significantly faster than both the penetration block and the average time required for a spermatozoon penetrating the PVS to fuse with the oolemma

To characterize the kinetics of the fusion block following the first fertilization event, we determined the temporal evolution of the average number of spermatozoa fused per oocyte after fertilization (dark blue dots in Fig. 5C). This number increases over time and eventually reaches a plateau. Among the 57 fertilized oocytes analyzed, three were dispermic, indicating that the fusion of a second spermatozoon can occasionally occur before the fusion block is fully established. The time evolution of the average number of fusions after a first fertilization was fitted with the function:

$$f(t) = \frac{3}{57}\left(1 - \exp\left(-\frac{t}{\tau_{FB}}\right)\right) \qquad (2)$$

The best fit yields a time constant $\tau_{FB} = 6.2 \pm 1.3$ minutes. In parallel, we determined the average number of available spermatozoa per oocyte in the PVS over time (purple dots in Fig. 5C). This analysis highlights the striking contrast between the 3 spermatozoa that successfully fused after a first fertilization (resulting in the 3 dispermic oocytes) and the many normally mobile spermatozoa present in the PVS that had penetrated the PVS but did not fuse. $\tau_{FB}$ can thus be interpreted as the time constant for the establishment of the fusion block after the first fertilization.

When compared to the time constant of the penetration block ($\tau_{PB} = 48.3 \pm 9.7$ min) and to the average time required for a spermatozoon to fuse with the oolemma after entering the PVS ($15.8 \pm 5.7$ min), the fusion block ($\tau_{FB} = 6.2 \pm 1.3$ min) emerges as significantly faster: approximately eight times faster than the penetration block and 2.5 times faster than the fusion time.

## Part of CD9 and JUNO released from the oolemma remain in the PVS for hours, where they bind to the sperm heads

Despite the key role of the fusion block in preventing polyspermy, its molecular bases are not elucidated yet. CD9 and JUNO, the two oocyte proteins identified to date as essential for fusion in mice (Bianchi et al, 2014; Kaji et al, 2000; Le Naour et al, 2000; Miyado et al, 2000), have been shown to undergo, following fertilization, partial and complete releases from the oolemma, respectively. These releases were hypothesized to contribute to the prevention of polyspermy (Bianchi et al, 2014; Ravaux et al, 2018). To proceed with this hypothesis, we performed standard in vitro fertilization

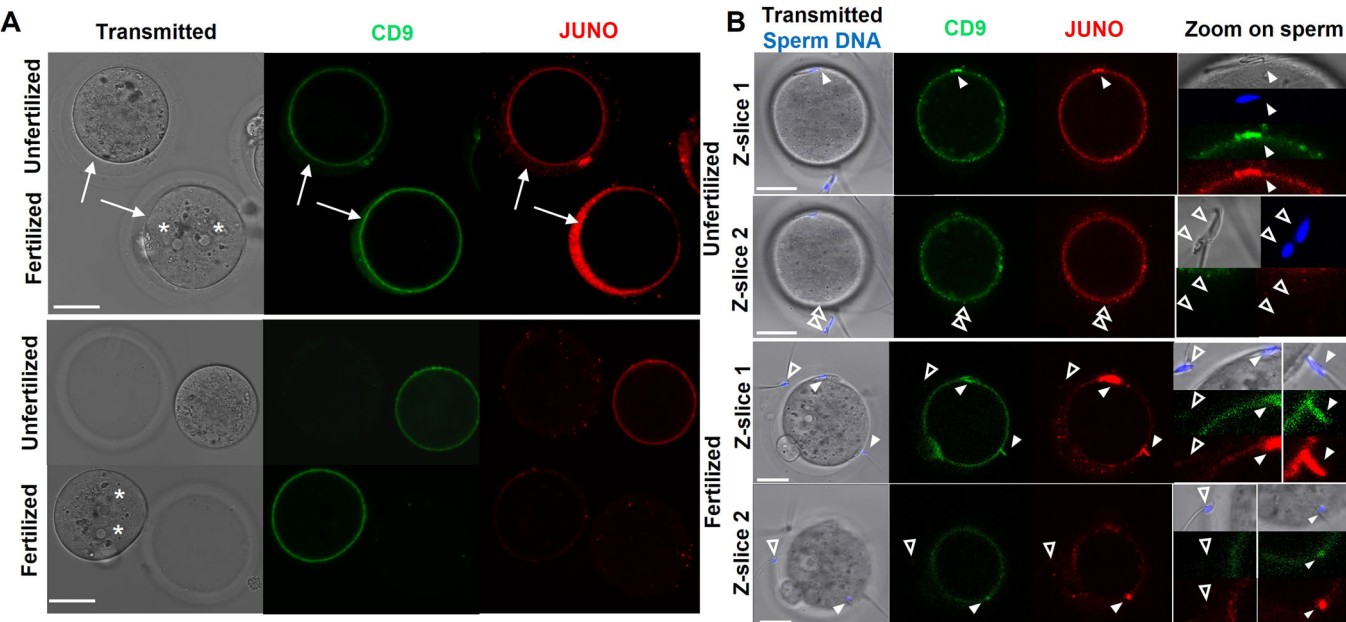

**Figure 6. Confocal images of CD9-EGFP oocytes stained with JUNO-dylight 633 antibody and Hoechst after 4 h of incubation with spermatozoa to assess the localization of the oocyte's CD9 and JUNO in the oocytes and on the spermatozoa in the PVS or bound to their ZP.**

(A) Unfertilized and fertilized oocytes before (top) and after (bottom) ZP removal. CD9 is localized mainly at the oolemma in the unfertilized oocyte, and both at the oolemma and in the PVS in the fertilized oocyte. JUNO is localized mainly at the oolemma in the unfertilized oocyte, and mainly in the PVS in fertilized. White arrows indicate PVS, stars indicate the two pronuclei in fertilized oocytes. (B) Unfertilized oocyte (top) and fertilized oocyte (bottom) with spermatozoa in the PVS (and bound to the ZP at two different z positions. The spermatozoa in the PVS (white filled arrow heads) are covered with both CD9 and JUNO, but not the sperm bound to the ZP (white empty arrow heads). For (A, B) Scales bars: 30 µm. Source data are available online for this figure.

assays with CD9-EGFP oocytes ($N = 30$), and studied the localization of CD9 and JUNO through confocal imaging after DNA and JUNO staining, typically 4 h after insemination. At the time of imaging, a majority of these oocytes were monospermic, fertilized for more than 3 h, as evidenced by the presence of two pronuclei in the ooplasm (Fig. 6A). However, some oocytes remained unfertilized (Fig. 6A). In both fertilized and unfertilized groups, a portion of oocytes had been penetrated by spermatozoa that failed to fertilize as attested by their presence in the PVS of these oocytes (Fig. 6B). Regarding the localization of CD9 and JUNO, we observe that before fertilization or in oocytes that remained unfertilized, CD9 and JUNO are predominantly localized at the oocyte membrane (Fig. 6A). In contrast, after fusion (four hours after insemination), CD9 is distributed between the membrane and the PVS of the oocyte, and the only JUNO signal detectable in the oocyte is found in the PVS (Fig. 6A). The fact that these molecules remain detectable in the PVS more than 3 h after fertilization (Fig. 6A) suggests that the ZP contributes to prevent the escape of CD9 and JUNO from the PVS after they have been released from the oolemma. We observed that the heads of all spermatozoa located in the PVS of the fertilized and unfertilized oocytes (white filled arrow heads in Fig. 6B) are coated with CD9 and JUNO, but not those bound to the outer surface of the ZP (white empty arrow heads in Fig. 6B). This observation raises the question of how the spermatozoa located in the PVS acquired these oocytes proteins. A key insight comes from the bottom right spermatozoon in Fig. 6B (fertilized z-slice 1), which was in the process of entering the PVS of a fertilized oocyte at the time of imaging. The spermatozoon head,

just emerged from the ZP and still oriented perpendicular to the oolemma—indicating that gamete adhesion has not yet begun—is already coated with CD9 and JUNO. This indicates that the spermatozoon acquired these proteins from those released into the PVS.

## Fertilization causes the de-adhesion of spermatozoa from the ZP

In parallel to the precise tracking of individual spermatozoa penetrating an oocyte, in vitro fertilization with kinetic tracking also enables some characterization of the evolution of the spermatozoa bound to the ZP over time. An approximate count of the number of spermatozoa bound to the oocyte ZP was possible for both fertilized oocytes and unfertilized oocytes with spermatozoa penetrating their PVS. At each round of observation, an oocyte was assigned a level from 1 to 4 based on the number of bound spermatozoa. Level 1 corresponds to fewer than 20 spermatozoa bound to the ZP, level 2 to 20 to 50 bound spermatozoa, level 3 to 50 and 100 bound spermatozoa, and level 4 when more than 100 spermatozoa were bound (Fig. 7A). Figure 7B, C illustrate the progression of the proportion of oocytes in each level over time, for the group of penetrated fertilized oocytes and the group of unfertilized oocytes, respectively. For the unfertilized oocytes, the proportion of oocytes in each level changes little over time, despite the entry of spermatozoa into the PVS. This shows that penetration does not significantly affect the capacity of the ZP to bind spermatozoa. For fertilized oocytes, they maintain their initial level

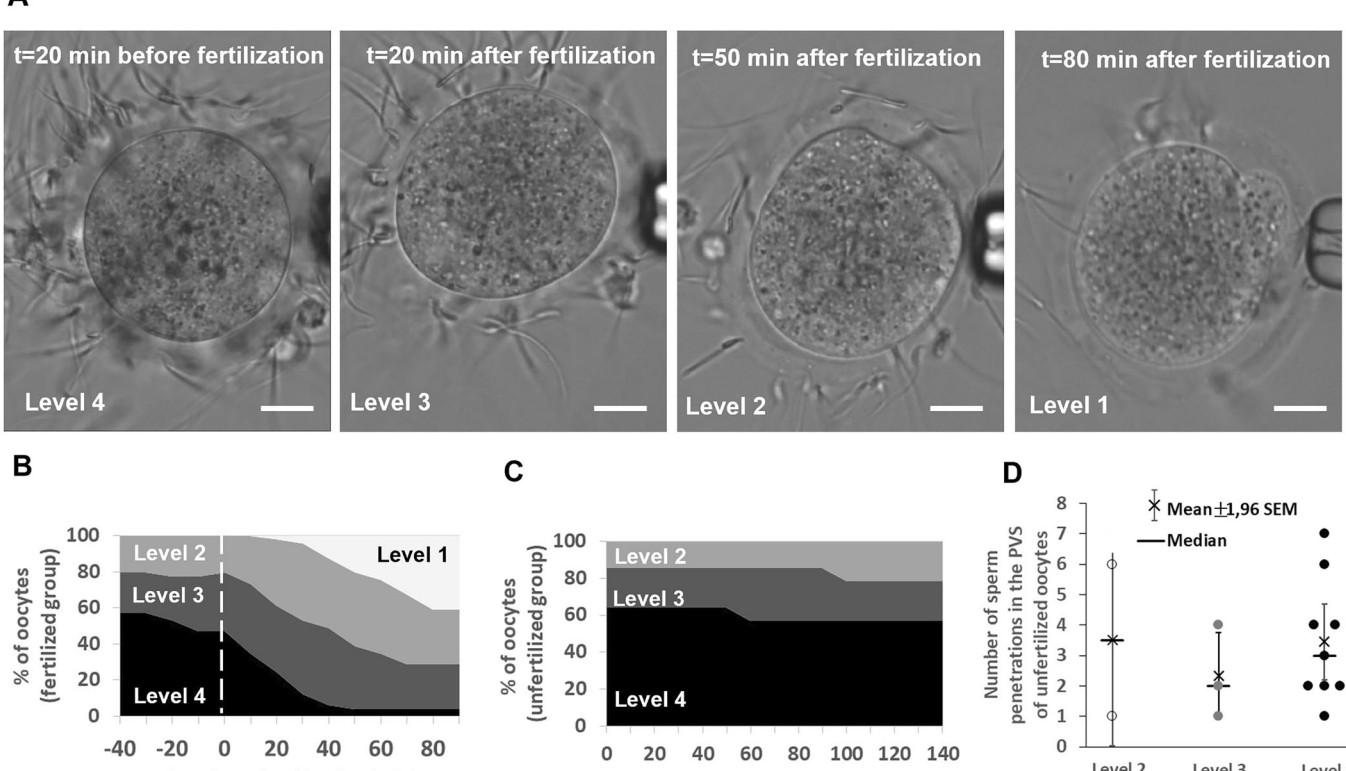

**Figure 7. Evolution of the number of spermatozoa bound to the ZP of an oocyte for fertilized and unfertilized oocytes.**

(A) Sequence of images of an oocyte before and after fertilization, illustrating the progressive de-adhesion of spermatozoa bound to its ZP after fertilization. Levels 4, 3, 2, and 1 refer to the range of spermatozoa bound to the ZP of an oocyte: Level 4 was assigned to an oocyte with more than 100 bound spermatozoa, Level 3 between 50 and 100 bound spermatozoa, Level 2 between 20 and 50 bound spermatozoa, and Level 1 less than 20 bound spermatozoa. Scales bars: 20 μm. (B) Temporal evolution of the proportion of oocytes in each level for the group of oocytes that remained unfertilized despite being penetrated. The dotted white vertical line marks the time point of fusion. (C) Temporal evolution of the proportion of oocytes associated with each level for the group of fertilized oocytes. (D) Number of spermatozoa penetrated in the PVS of unfertilized oocytes as a function of the number of spermatozoa bound to their ZP. Source data are available online for this figure.

until fertilization occurs, after which they progressively shift to lower levels as spermatozoa detach. This suggests that fusion, unlike penetration, triggers a loss of affinity of the ZP for spermatozoa. Despite the poor accuracy in the quantification of the ZP de-adhesion kinetics compared to the ZP permeability inhibition kinetics, we can reasonably say that both permeability and adhesion inhibitions of the ZP occur within the same timeframe (~1 h). This raises the question of whether loss of adhesion and loss of permeability are two independent manifestations of a fertilization-triggered process, or whether loss of penetrability is directly the consequence of the decrease in ZP-bound spermatozoa due to de-adhesion. The fact that the group of penetrated but unfertilized oocytes shows no correlation between the number of ZP-bound spermatozoa and the occurrence of penetrations argues in favor of two distinct manifestations (Fig. 7D).

## Discussion

### Fertilization failure of the spermatozoa traversing the ZP of unfertilized oocytes may result from inappropriate

**flagellum beating or from their too longer kinetics of fertilization as compared to the fusion block kinetics**

The chronogram of the 57 fertilized oocytes reveals that it is not uncommon for the first spermatozoon that enters the PVS of an oocyte and interacts with the oolemma to fail in fertilizing it. It is also not uncommon for two or even three spermatozoa to enter the PVS of an oocyte before one of them succeeds in fertilizing it. Although this situation presents the highest risk of polyspermy, polyspermy occurs only rarely in mice. Yet, when these spermatozoa entered the PVS, the oocyte was in its optimal state for fertilization, as it had not yet been fertilized and, therefore, no fertilization-triggered blocking mechanisms had been activated. Moreover, all the spermatozoa that successfully reach the PVS of an oocyte have undergone the acrosome reaction (Jin et al, 2011; Jabloñski et al, 2024), which is the process during which the ultimate modifications that make the spermatozoon capable of fertilizing an oocyte occur (Yanagimachi, 1994). The reason why these spermatozoa failed to fertilize is therefore intriguing. Analyzing the choreography of these spermatozoa revealed two possible explanations. One is the intrinsic inability of certain spermatozoa to fertilize once their flagellum is fully inside the PVS, due to flagellar beating that causes their head to interact with the

oolemma in a way that is incompatible with fusion. The second possibility concerns the delay between the emergence of the head from the ZP into the PVS and its fusion with the oolemma. We observed that this process involves three phases, and that only the final phase, characterized by push-up-like movements of the spermatozoon head against the oolemma, can lead to fusion. Each phase lasts a few minutes, with a total average time of $15.8 \pm 5.7$ min. This is typically more than twice as long as the $6.2 \pm 1.3$ min time constant determined for the fusion block. This suggests that any spermatozoon that has penetrated the PVS a few minutes before or after the fertilization of the oocyte by another spermatozoon is likely to have its chances of fusion thwarted by the faster fusion block, before it could have reached the push-up-like phase leading to fusion. Another possible explanation for the failure of spermatozoa that have traversed the zona pellucida of unfertilized oocytes to fuse with the oolemma arises from a recent study showing the existence of two distinct populations of live, acrosome-reacted spermatozoa(Jabloñski et al, 2024). These correspond to two successive stages, which occur either immediately upon acrosome reaction in a subset of spermatozoa, or after a variable delay in others, during which the sperm transitions from a motile to an immotile state. The transition from the first to the second stage was shown to follow a defined sequence: an increase in calcium concentration in the flagellar midpiece, followed by midpiece contraction associated with a local reorganization of the helical actin cortex, and ultimately the arrest of sperm motility. For fertilizing spermatozoa in the PVS, this transition was consistently observed upon fusion (Jabloñski et al, 2024). However, it was also reported in some non-fertilizing spermatozoa that this transition took place within the PVS. These findings are consistent with the requirement for sperm motility in order to achieve fusion with the oolemma. Moreover, the fact that some spermatozoa may prematurely transition to the immotile state within the PVS can therefore be added to the list of possible reasons why a spermatozoon that penetrates the PVS of an oocyte might fail to fuse.

## New mechanical role of the ZP: promoting gamete fusion by channeling flagellar movements

The correlation between flagellum beating and gamete fusion was first demonstrated in a previous study performed with ZP-free oocytes (Ravaux et al, 2016), where the only limitation to flagellum movement was provided by the adhesion of the head to the oolemma. Here, our observation with physiological ZP-intact oocytes allows us to identify the same beating patterns— vigorous or erratic—in spermatozoa whose flagellum manages to fully extract itself from the ZP, as those observed with ZP-free oocytes. We also observe the same consequences on their head movements on the oolemma—push-up-like or abrupt head rotations tangential to the oolemma—and the same correlations with fertilization success or failure. This suggests that the PVS, although a narrow space, remains large enough not to restrict flagellum movements in a way that could change sperm ability or inability to fertilize. By contrast, the entrapment of the flagellum within the ZP imposes strong restrictions. This makes the choreography of all spermatozoa look similar, making it impossible to predict whether their beating pattern would be compatible with fusion if they were to fully penetrate the PVS before undergoing fusion. As long as spermatozoa are not fully released from their entrapment in the ZP, the constraints imposed by this entrapment regulate their

movement, giving all of them a chance to fertilize before these constraints are lifted. This reveals a previously unsuspected function of the ZP in facilitating fertilization by channeling flagellum beating.

## In mice, the prevention of polyspermy relies predominantly on the fusion block, supported by the naturally low permeability of the ZP and the relatively slow fusion kinetics, whereas the too slow penetration block plays only a marginal role

Our statistical analysis of the experimentally determined time windows for sperm penetration and fusion allowed us to extract the average kinetics of the four following factors: (i) the fusion time after sperm entry in the PVS: we determined that, on average, $15.8 \pm 5.7$ min elapse between the entry of a fertilizing spermatozoon into the PVS and its fusion with the oolemma, (ii) the baseline permeability of the ZP permits the penetration in the PVS of $1.21 \pm 0.37$ spermatozoa per hour per oocyte before fertilization, (iii) after the first fertilization, a penetration block is established with a time constant of $48.3 \pm 9.7$ min, and (iv) a fusion block is set with a time constant of $6.2 \pm 1.3$ min. To assess the relative impact of these factors (fusion time, ZP natural baseline permeability, ZP penetration block and fusion block of spermatozoa in the PVS), we have used our experimentally determined kinetics of these factors to make a kinetic numerical Monte Carlo simulation of spermatozoa penetrating the PVS and fusing with the oocytes at realistic rates. As shown by the blue curve in Fig. 8, this simulation accurately reproduces the observed temporal evolution of the average number of spermatozoa fusing with an oocyte after a first fertilization (black dots). Interestingly, this numerical approach can be used to selectively turn off the fusion block (green curve), the penetration block (orange curve), to adjust the natural baseline permeability of the ZP (red curve) or the average fusion kinetics (purple curve), thereby enabling a quantitative assessment of the contributions of each factor in preventing polyspermy (Fig. 8). From these simulations, we found that the fusion block plays a critical role: its absence leads to a dramatic increase in polyspermy. The speed at which a fertilizing spermatozoon fuses after entering the PVS also plays a key role: faster fusion leaves less time for the fusion block to prevent spermatozoa that entered the PVS just before or just after fertilization from also fusing, thus increasing the risk of polyspermy. Similarly, the ZP baseline permeability influences polyspermy risk because higher permeability leads to more spermatozoa in the PVS prior to or shortly after fertilization when the fusion block is still not fully efficient. In contrast, the penetration block contributes marginally to preventing polyspermy, as it becomes effective long after the fusion block. This is in agreement with the finding that mice missing ovastacin -the protease that renders the ZP impenetrable- are normally fertile and explains why, after mating, the oocytes of these mice remain monospermic despite the systematic presence of additional spermatozoa in their PVS (Nozawa et al, 2018).

## Relation between the penetration block and the ZP-block

This study shows that the penetration block taking place after a first fertilization is characterized by a progressive and sustained loss of both ZP permeability to spermatozoa and its capacity to bind them.

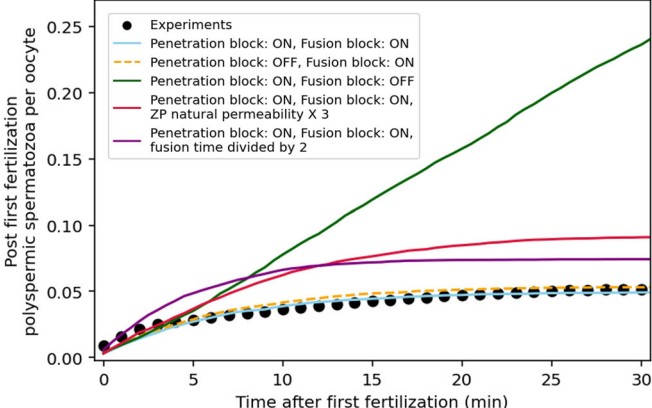

**Figure 8. Monte Carlo simulations to assess the relative impact of the natural baseline permeability of the ZP, the ZP penetration block and the fusion block in preventing polyspermy.**

The black dots correspond to the experimental temporal evolution of the average number of spermatozoa fusing with an oocyte after a first fertilization and thus resulting in polyspermy. Monte Carlo simulations of sperm penetrations in the PVS and fusions, using the kinetic parameters experimentally obtained for (i) the time for a fertilizing spermatozoon to fuse once penetrated in the PVS (on average 15.8 min) Fig. 5A), (ii) the natural baseline permeability of the ZP (on average 1.2 spermatozoa penetrating the PVS per hour per oocyte (Fig. 4B; Appendix Fig. S1B)), (iii) the penetration block (48.3 min time constant) (Fig. 5B)), (iv) the fusion block (6.2 min time constant (Fig. 5C)). Simulations run without ZP penetration block (orange curve), without fusion block (green curve), with tripled natural ZP permeability (red curve), and half fusion time (purple curve) emphasize the impact of the different factors on polyspermy. Source data are available online for this figure.

These two changes in ZP properties occur within a similar timeframe of ~1 h (Figs. 3B and 7A,B). The fertilization-triggered processes responsible for the changes in the ZP properties are generally attributed to the cortical reaction—a calcium-induced exocytosis of secretory organelles (cortical granules) initially present in the cortex of unfertilized mammalian oocytes—and to zinc sparks (Austin, 1956; Gulyas, 1980; Liu, 2011; Que et al, 2017). As a result, proteases, glycosidases, lectins and zinc are released in the PVS and act on ZP components, leading to a series of modifications collectively referred to as ZP hardening or ZP-block. These changes occur over a period ranging from 30 min to several hours, during which the ZP progressively loses its ability to bind to and be penetrated by spermatozoa (Baibakov et al, 2007; Barros and Yanagimachi, 1971; Bleil et al, 1981; Burkart et al, 2012; Fahrenkamp et al, 2020; Miller et al, 1993; Nishio et al, 2024; Que et al, 2017; Tokuhiro and Dean, 2018). The de-adhesion of spermatozoa from the ZP after fertilization and the progressive decrease in ZP permeability we have observed in this study are consistent with the timing and phenotypes of the ZP-block ascribed to the cortical reaction. It is therefore likely that the ZP-block alone accounts for the observed penetration block.

## Relation between the fusion block and the membrane- and PVS-blocks

The most extensively documented component of the fusion block is the so-called membrane-block, which relies on functional changes in the oolemma after fertilization that render it less receptive to spermatozoa. In vitro insemination experiments using ZP-free oocytes from various mammalian species—including human, mouse, hamster, and rabbit—have specifically investigated the phenotypes associated with the membrane-block. These studies revealed that the fusion of a spermatozoon with the oolemma of an unfertilized oocyte triggers a progressive and persistent loss of the oolemma's ability to bind to and fuse with additional spermatozoa (Wolf, 1978; McCulloh et al, 1987; Zuccotti et al, 1991; Sengoku et al, 1995; Maleszewski et al, 1996; McAvey et al, 2002; Gardner et al, 2007; Kryzak et al, 2013; Evans, 2020). In mice, this membrane-block has been reported to be fully established within ~1 h after fertilization (Gardner et al, 2007; Kryzak et al, 2013; Maluchnik and Borsuk, 1994; Wolf, 1978). Although the membrane-block's hallmark loss of oolemma adhesion and fusion capacity is consistent with our observation of poor adhesion between the oolemma of fertilized oocytes and penetrating spermatozoa, the timing at which the oolemma becomes fusion-incompetent does not align with the faster fusion block kinetics we observed. This strongly suggests that the membrane-block alone cannot fully account for the fusion block.

The second component of the fusion block is associated with the PVS and is referred to as the PVS-block (Talbot and Dandekar, 2003). Its existence has been inferred from in vitro fertilization assays conducted in mice and rabbits (McCulloh et al, 1987; Maluchnik and Borsuk, 1994). These studies showed that ZP-intact oocytes rarely become polyspermic, even when numerous additional spermatozoa (up to a dozen in mice and up to more than a hundred in rabbits) are present within the PVS. In contrast, the incidence of polyspermy increases significantly when the same oocytes are inseminated with fresh spermatozoa following zona pellucida removal. This additional contribution to the fusion block prevents the spermatozoa in the PVS from fertilizing the oocyte while the binding and fusion properties of the oolemma are still functional. This supports the existence of a PVS-block operating with faster kinetics than the membrane-block, consistent with the fusion block kinetics we observed. These findings raise the possibility that the fusion block and, therefore, prevention of polyspermy mainly rely on this PVS-block in mice.

## Possible molecular bases of the membrane-block and PVS-block contributing to the fusion block

Despite the crucial role of the fusion block in preventing polyspermy in mice—and likely in several other mammalian species—its molecular bases remain unknown. The hypothesis that the fertilization-induced cortical reaction, which is responsible for the ZP-block, might also underlie the membrane- and PVS- blocks contributing to the fusion block was investigated and ultimately ruled out nearly two decades ago (Dandekar and Talbot, 1992; Horvath et al, 1993; Hoodbhoy et al, 2001; Talbot and Dandekar, 2003; Gardner et al, 2007; Evans, 2020). Since then, research into the molecular mechanisms underpinning the membrane- and PVS-blocks has largely stagnated. However, it is now known that following fertilization, the zygote releases oocyte membrane proteins such as JUNO and CD9 into the PVS (Bianchi et al, 2014; Ravaux et al, 2018), opening new possibilities for investigating the molecular players involved in the fusion block.

JUNO, the oocyte receptor for the sperm protein IZUMO1, is essential for sperm–oocyte adhesion leading to fusion; its shedding

from the oolemma after fertilization has been proposed as a possible mechanism for eliminating sperm-binding competence at the surface of the fertilized oocyte, a defining feature of the membrane-block (Bianchi et al, 2014). Similarly, CD9, the second oocyte protein known to be essential for successful sperm–oocyte interaction leading to fusion, is partially released into the PVS within minutes of gamete fusion (Ravaux et al, 2018). The presence of these two proteins in the PVS suggests that they may not simply be passive byproducts of membrane remodeling but could play an active role.

In the 2000s, a model has been proposed in which CD9-containing vesicles, already released by unfertilized oocytes, mediate gamete fusion by interacting with spermatozoa. This was supported by observations showing that spermatozoa, when incubated with perivitelline content from wild-type oocytes, became coated with CD9-bearing vesicles and subsequently gained the ability to fuse with CD9-deficient ZP-free oocytes (Miyado et al, 2008). However, this model was later called into question, as two independent studies failed to reproduce these findings under comparable experimental conditions (Gupta et al, 2009; Barraud-Lange et al, 2012).

Here, our observations of non-fertilizing spermatozoa being coated by JUNO and CD9 in the PVS lead us to propose a fundamentally different hypothesis: rather than facilitating fusion, shed JUNO and CD9 may contribute to the fusion block by binding to spermatozoa in the PVS and competitively inhibiting further interactions with the oolemma. According to this hypothesis, the PVS-block contributing to the fusion block would result from the neutralization of spermatozoa by impairing their ability to properly adhere to and fuse with the oolemma.

Regarding the respective kinetics of membrane- and PVS-blocks, although both are likely triggered by the same process - the release of oolemma molecules including JUNO and CD9 following fertilization- they may not operate on the same time scale. Based on previous studies using zona-free oocytes to assess the timing of loss of oolemma receptivity to spermatozoa, the membrane-block appears to become fully effective, typically one hour after fertilization (Kryzak et al, 2013; Maluchnik and Borsuk, 1994). This timing coincides with the complete clearance of JUNO from the oolemma (Bianchi et al, 2014). According to the hypothesis that the membrane block depends on the removal of JUNO from the oolemma, these data suggest that almost complete JUNO depletion is required for the membrane-block to be fully established. In contrast, we expect that the PVS-block requires only partial release of CD9 and JUNO, achievable within a few minutes, for complete passivation of the penetrating spermatozoa.

In summary, this study sheds new light on the complex mechanisms that enable fertilization and ensure monospermy in the mouse model which opens new perspectives for future research. A crucial step will be to elucidate how the mechanical constraints exerted by the spermatozoon head on the oolemma contribute to create a local physico-chemical environment conducive to fusion. Another crucial next step will be to test experimentally whether JUNO and CD9 directly inhibit sperm fusion competence after their release into the PVS. Understanding the molecular form and functional state of these proteins in the PVS, and the mechanism by which they bind to and potentially inactivate sperm, could significantly advance our knowledge of the events preventing polyspermy. In parallel, comparative studies in

other mammalian species will be needed to assess the generality of the PVS-block and its contribution relative to the membrane-block and ZP-block, as well as the generality of the mechanical role played by flagellar beating and ZP mechanical constraint in promoting membrane fusion. Finally, these insights may have translational implications in assisted reproductive technologies and contraception, offering new targets for modulating fertilization efficiency or blocking sperm–oocyte interactions in a specific manner.

## Methods

**Reagents and tools table**

| Reagent/resource | Reference or source | Identifier or catalog number |
|---|---|---|
| **Experimental models** | | |
| B6cbaF1/J (*M.musculus*) | Charles River France | 631B6cbaF1 (JAC ® Mice strain) |
| CD9-EGFP (*M.musculus*) | Kenji Miyado (national research institute for child health and development), Tokyo, Japan. Miyado et al, 2008 | C57BL/6-Cd9tm1OsbTg(ZP3-EGFP/Cd9) |
| **Hormones** | | |
| Gonadotropine sérique (PMSG) | CHRONO.GEST® PMSG 600 | |
| Gonadotropine chorionique (hCG) | CHORULON® 1500 | |
| **Culture medium** | | |
| M2 medium with Hepes | Sigma-Adrich | M7167-50ML |
| Ferticult IVF medium | JCD Laboratories | MT246-5 |
| **Antibodies** | | |
| Rat antibody against mouse-JUNO (anti-FR4) | Miltenyi Biotec | 130-095-244 |
| **Other reagents** | | |
| Bovine serum albumin, Powder, Bioxtra | Sigma-Adrich | A3311-50G |
| Hoechst 33342 | Thermo Fisher France | H3570 |
| DyLight ™ 633 NHS Ester antibody labeling kit | Thermo Fisher Scientific | 53046 |
| Mineral oil | JCD Laboratories | MT260 |
| **Tools** | | |
| Confocal SP5 | Leica | N/A |
| Camera UI-3040SE Monochrome | IDS | N/A |
| **Software** | | |
| LAS X | Leica | N/A |
| uEye Cockpit | IDS | N/A |
| ImageJ | https://imagej.nih.gov/ij/index.html | N/A |

| Reagent/resource | Reference or source | Identifier or catalog number |
|---|---|---|
| Spyder for Python | https://www.spyder-ide.org/download | N/A |

## Ethics statement

All animal experiments were performed in accordance with national guidelines for the care and use of laboratory animals. Authorizations were obtained from local (Animal Care and Use Committee Charles Darwin, France (#30204)) and governmental ethical review committees via APAFIS Application, authorization APAFIS #30204-2021030415199914 v3.

## In vivo fertilization (Condition 1) and determination of the number of penetrating and fertilizing spermatozoa

Eight to 12-week-old wild-type (WT) female mice (B6cbaF1/J background) were superovulated by intraperitoneal injections, first of 5 IU Gonadotropine sérique (PMSG), followed by 5 IU Gonadotropine chorionique (hCG) 48 h apart and immediately mated with a male. Fifteen hours after mating, plugged females were sacrificed, oocytes were recovered, washed, fixed and stained with Hoechst and imaged in confocal microscopy to determine the number of penetrations and fertilizations of each oocyte. Ten independent in vivo experiments were performed for a total of 211 oocytes.

## Sperm preparation for in vitro fertilization experiments

Sperm were obtained from WT male mice (B6cbaF1/J background). Sperm were expelled from the cauda epididymis and vas deferens into Ferticult® IVF medium under mineral oil. Sperm were then incubated in Ferticult® at 37 °C, 5% $CO_2$ in air for 1.5 h to induce capacitation.

## Oocyte preparation for in vitro fertilization experiments

*CD9-EGFP* and *WT* oocytes were obtained from 6 to 12-week-old female mice (B6cbaF1/J background for WT mice and C57BL/6 J background for CD9-EGFP transgenic mice oocytes (Miyado et al, 2008)). Female mice were superovulated as previously described. Cumulus–oocyte complexes were collected into a Ferticult® IVF medium drop 14 h later by tearing the oviduct ampulla from sacrificed mice.

## Standard in vitro fertilization (Condition 2) and in vitro fertilization with kinetic tracking (Condition 3) protocols

Both kinetic and standard in vitro fertilization experiments were initiated by inseminating in a 200 μL drop of Ferticult the four cumulus–oocyte complexes, retrieved from two superovulated females, with capacitated spermatozoa at a concentration of $10^6$ spermatozoa/mL. The inseminated cumulus–oocyte complexes were then incubated at 37 °C in an atmosphere containing 5% $CO_2$. An incubation period of approximately 15 min was generally sufficient for most cumulus cells to detach from the oocytes, and for the majority of oocytes to become surrounded by numerous spermatozoa (typically a few dozens) bound to their zona pellucida (ZP), without any observable sperm penetration or fertilization at this stage.

In the in vitro fertilization experiments with kinetic tracking, a subset of oocytes (in average 5 oocytes per experiment)—together with their associated ZP-bound spermatozoa—were isolated 15 min after insemination and transferred individually into microdrops of fertilization medium to enable identification. These oocytes were gently immobilized using a pipette and observed sequentially at high magnification over a period of ~4 h. The duration of each observational sequence could vary from less than one minute when no sperm penetration into the PVS or interaction with the oolemma was in progress, to typically around 10 min when such events were in progress. Consequently, as in average five oocytes were observed in turn, the interval between two rounds of observation of a given oocyte could range from a few minutes to more than an hour.

In parallel, standard in vitro fertilization experiments were conducted by maintaining the remaining oocytes in co-incubation with spermatozoa for 4 h. Both kinetic and standard experiments were terminated 4 h after insemination. At that point, oocytes were washed to remove any spermatozoa still adhering to the ZP, fixed and subjected to confocal microscopy following DNA staining. A total of 11 independent in vitro experiments were conducted, resulting in 93 oocytes analyzed with kinetic tracking and 220 oocytes subjected to standard in vitro fertilization protocols.

## Determination of the penetrating and fertilizing status of spermatozoa during in vitro fertilization with kinetic tracking experiments

The few oocytes isolated 15 min after insemination with the spermatozoa attached to their ZP from the insemination chamber were imaged in bright field in turns at 37 °C, by successively immobilizing them at the tip of a micropipette through gentle aspiration. When immobilized, the whole volume of the oocyte was explored by tuning vertically the objective (and therefore the focus plane) in order to screen plane by plane the oocyte and associated spermatozoa, looking for penetrating and fusing sperm. Triple checking was done by changing twice the position of the oocyte at the tip of the pipette in order to obtain three different perspectives of the oocyte. The numbers of penetrated and fused spermatozoa were counted at each round of observation of an oocyte and compared to the previous round, allowing to determine the order of penetration of each spermatozoon into the PVS of a given oocyte and its fertilization status. A spermatozoon was considered penetrated when at least its head was emerged from ZP into the PVS. A spermatozoon was further considered fused when its flagellum was immobile and its head lying on the oolemma or partially or totally engulfed in the oocyte.

## Determination of the penetration time windows (Fig. 3A and Appendix Fig. S1A)

A spermatozoon was considered to have penetrated when its head fully emerged from the ZP. During in vitro fertilization experiments with kinetic tracking, each oocyte is observed sequentially in turn rather than continuously. As a result, sperm penetration into the PVS of a given oocyte may occur either during an observation round or in the interval between two rounds. In the former case,

penetration is directly observed in real-time, allowing for high temporal precision in determining the exact moment of penetration. In contrast, when penetration occurs between two observation rounds (e.g., between rounds N–1 and N), the precise time of penetration cannot be directly determined. Instead, only an estimated penetration time window—defined by the interval between the end of round N–1 and the beginning of round N—can be established. Sperm penetration is assumed to have occurred at some point within this interval. In Fig. 3A; Appendix Fig. S1A, these estimated penetration time windows are represented by dashed black segments, scaled to reflect their respective duration.

## Determination of the fertilization time windows (Fig. 3A)

In a previous study[4], we investigated the temporal relationship between the abrupt cessation of sperm head movement on the oolemma—resulting from strong flagellar beating arrest—and the fusion event, using ZP-free oocytes preloaded with Hoechst. That study revealed a temporal delay of less than one minute between the cessation of sperm oscillations and the fusion event, thereby supporting the conclusion that in ZP-free oocytes, the arrest of vigorous sperm movement at the oolemma is a reliable indicator of the moment at which fusion occurs. In the same study, the kinetics of sperm head internalization into the ooplasm were also characterized, typically concluding within 20–30 min after movement cessation. These findings are fully consistent with our current observations in ZP-intact oocytes, where sperm head engulfment was completed approximately $24 \pm 3$ min after the arrest of sperm head push-up-like oscillations (Appendix Fig. S2C). Taken together, these results strongly support the conclusion that, in both ZP-free and ZP-intact oocytes, the arrest of sperm head push-up-like movement is a reliable indicator of the fusion event. This assumption formed the basis for our determination of fertilization time points in the present study.

Accordingly, when this sudden cessation of sperm movement was observed during an observation round and was subsequently followed by sperm head internalization, the time of movement arrest $\pm 1$ min was taken as the fusion time window. Conversely, when fusion occurred between two observation rounds (e.g., between rounds N–1 and N), the interval between the end of round N–1 and the beginning of round N was taken as a first determination of the fusion time window. However, this first estimate was further refined using additional temporal landmarks based on the kinetics of sperm engulfment and subsequent PB2 extrusion, as quantified in Appendix (Appendix Fig. S2C,D). These landmarks are as follows:

(i)   the disappearance of the sperm head from the oolemma due to internalization is completed $24 \pm 3$ min (mean $\pm$ SD) after fusion,
(ii)  the onset of PB2 protrusion from the oolemma takes place $28 \pm 2$ min after fusion
(iii) the contact angle between the PB2 protrusion and the oolemma shifts from greater than to less than 90° typically $49 \pm 6$ min after fusion
(iv)  PB2 extrusion is completed $73 \pm 10$ min after fusion

- If at insemination time of observation $t_{obs}$ after insemination the head of the fused spermatozoon was still visible at the oolemma, then according to landmark (i), fusion was considered to have taken place at time $t_F$ within the time window: $t_{obs} - 27$ min $< t_F < t_{obs}$
- If only the sperm flagellum but not the head of the fused spermatozoon was still visible, then, according to landmark (i), fusion was considered to have taken place at: $t_F < t_{obs} - 21$ min
- If a protrusion corresponding to PB2 was observed and its contact angle with the oolemma was higher than 90°, then according to landmark (ii) and (iii) fusion was considered to have taken place at time $t_F$ within the time window: $t_{obs} - 55$ min $< t_F < t_{obs} - 26$ min
- If PB2 was still in the extraction process and its contact angle with the oolemma was already smaller than 90°, then according to landmark (iii) and (iv) fusion was considered to have taken place at time $t_F$ within the time window: $t_{obs} - 83$ min $< t_F < t_{obs} - 43$ min
- If the PB2 was observed completed at $t_{obs}$, then according to landmark (iv) fusion was considered to have taken place at $t_F < t_{obs} - 63$ min

Since an oocyte could be observed prior to fertilization, during the interval between the four temporal landmarks, or after the completion of PB2 extrusion, up to six independent estimations of the fusion time window could be obtained for a single oocyte. Provided that the state of each oocyte was correctly identified during each observation round, all of these time windows should overlap—which was indeed the case—yielding a common interval that corresponds to the most accurate estimation of the fusion time window. In Fig. 3A, these fertilization time windows are represented by solid black segments, scaled to reflect their respective duration.

## Statistical treatment of penetrations and fertilizations chronograms to study the penetration and fusion block kinetics

Our experiments allow the determination of time windows for each penetration and fertilization event. To estimate the average number of fertilization or penetration or polyspermic events per oocyte, as well as their confidence intervals, as shown in Fig. 5, we implemented a statistical analysis based on the simple assumption that each event takes place at a given time in its time window with a uniform probability distribution. The rationale behind this assumption is that, since we can only ascertain that the event took place within this time window, there is no reason to favor one time point over another. Based on this assumption, we numerically computed the probability distributions of the number of spermatozoa that penetrated or fertilized for each oocyte, using timescales aligned either to the first penetration event (Appendix Fig. S1B) or to the first fertilization event (Fig. 5B,C). From these distributions, we derived the average number of penetrating and fertilizing spermatozoa per oocyte. The Python scripts used to perform these calculations are available in the source data file provided for Fig. 5A–C.

## Determination of the number of penetrating and fertilizing spermatozoa after standard in vitro fertilization experiments and in vitro fertilization with kinetic tracking

Four hours after insemination, all oocytes from both standard and kinetic in vitro fertilization experiments were washed to remove

spermatozoa still bound to the ZP, fixed, stained with Hoechst, and imaged using confocal microscopy. For each oocyte, the number of spermatozoa that had successfully traversed the ZP was assessed by counting the distinct DNA signals corresponding to sperm nuclei in the PVS, indicative of penetration without fertilization. Fertilizing spermatozoa were identified based on the presence of decondensed sperm nuclei and/or male pronuclei within the oocyte cytoplasm. Notably, for the kinetic in vitro fertilization experiments, this post-hoc analysis confirmed the real-time identifications made during the 4-h live imaging, thereby validating our ability to accurately distinguish, under brightfield microscopy, spermatozoa entering the PVS from those merely attached to the ZP.

## Localization of CD9 and JUNO by confocal imaging of CD9-EGFP oocytes and penetrated spermatozoa

Four hours after insemination of cumulus-intact CD9-EGFP oocyte masses with spermatozoa ($10^6$ spermatozoa/mL), the oocytes were washed to free them from bound spermatozoa. The oocytes were loaded with Hoechst for DNA staining and stained with anti-JUNO/FOLR4 monoclonal antibody (BioLegend) coupled with Dylight 633. Samples were imaged with a Leica SP5 confocal microscope on the lookout for the localization of CD9 and JUNO at the oolemma, in the PVS and on the penetrated spermatozoa.

# Data availability

This kinetic studies of fertilization and polyspermy prevention is based on the analysis of recordings of inseminated oocytes observed individually. The recordings used for this study, as well as the confocal images that retrospectively validate our ability to identify in real time the spermatozoa penetrating the perivitelline space (PVS), interacting with the oolemma, and fusing with the oolemma, have been deposited in the BioImage Archive database https://www.ebi.ac.uk/biostudies/bioimages/studies/S-BIAD2356 and assigned the identifier S-BIAD2356. The analysis of these recordings provided the source datasets for Figs. 3A,–C, 5A–C, 7B–D, and 8; Appendix Figs. S1A,B and S2C,D from the Appendix. Movies EV1–EV5 are selected excerpts from these recordings.

The source data of this paper are collected in the following database record: biostudies:S-SCDT-10_1038-S44319-025-00670-8.

# Peer review information

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

## Acknowledgements

We warmly thank Gwendoline Firmin, Eleonore Touzalin, and Amandine Delecourt from IBENS for technical help with mice. We thank Sophie Cribier from LBM and Frédéric Pincet from LPENS for fruitful discussions. This work was supported by the Agence Nationale pour la Recherche ANR-21-CE13-0032 FUSOGAME grant.

## Author contributions

**Yaëlle Dubois**: Conceptualization; Formal analysis; Investigation; Visualization; Methodology; Writing—original draft. **Sophie Favier**: Conceptualization; Investigation; Visualization; Methodology. **Nathan Martin-Fornier**: Formal analysis; Visualization; Methodology. **Adrien Freyss**: Investigation; Visualization. **Mohyeddine Omrane**: Investigation. **David Stroebel**: Supervision; Validation. **Eric Perez**: Supervision; Validation. **Sandrine Barbaux**: Supervision; Funding acquisition; Validation. **Ahmed Ziyyat**: Supervision; Funding acquisition; Validation. **Nicolas Rodriguez**: Conceptualization; Data curation; Formal analysis; Supervision; Validation; Investigation; Visualization; Methodology; Writing—original draft; Project administration; Writing—review and editing. **Christine Gourier**: Conceptualization; Data curation; Formal analysis; Supervision; Funding acquisition; Validation; Investigation; Visualization; Methodology; Writing—original draft; Project administration; Writing—review and editing.

Source data underlying figure panels in this paper may have individual authorship assigned. Where available, figure panel/source data authorship is listed in the following database record: biostudies:S-SCDT-10_1038-S44319-025-00670-8.

## Disclosure and competing interests statement

The authors declare no competing interests.

