## [Peer Review File · EMBO Reports]

Roles of the zona pellucida in gamete fusion and of the perivitelline space in blocking polyspermy in mice

Yaëlle Dubois, Sophie Favier, Nathan Martin-Fornier, Adrien Freyss, Mohyeddine Omrane, David Stroebel, Eric Perez, Sandrine Barbaux, Ahmed Ziyat, Nicolas Rodriguez, and Christine Gourier

Corresponding author(s): Christine Gourier (christine.gourier@ens.fr) , Nicolas Rodriguez (nicolas.rodriguez@ens.fr)

Review Timeline:

Submission Date:	9th Jul 25
Editorial Decision:	10th Sep 25
Revision Received:	23rd Oct 25
Editorial Decision:	13th Nov 25
Revision Received:	18th Nov 25
Accepted:	26th Nov 25

Editor: Martina Rembold / Esther Schnapp

**Transaction Report: This manuscript was transferred to
EMBO reports following peer review at Review Commons.**

**Review
COMMONS**

Review #1

1. Evidence, reproducibility and clarity:

Evidence, reproducibility and clarity (Required)

The manuscript "Key roles of the zona pellucida and perivitelline space in promoting gamete fusion and fast block to polyspermy inferred from the choreography of spermatozoa in mice oocytes" by Dr. Gourier and colleagues explores the poorly understood process of gamete fusion and the subsequent block to polyspermy by live-cell imaging of mouse oocytes with intact zona pellucida in vitro. The new component in this study is the presence of the ZP, which in prior studies of live-cell imaging had been removed before. This allowed the authors to examine contributions of the ZP to the block in polyspermy in relation to the timing of sperm penetrating the ZP and sperm fusing with the oocyte. By carefully analysing the timing of the cascade of events, the authors find that the first sperm that reaches the membrane of the mouse oocyte is not necessarily the one that fertilizes the oocytes, revealing that other mechanisms post-ZP-penetration influence the success of individual sperm. While the rate of ZP penetration remains constant in unfertilized oocytes, it decreases upon fertilization for subsequent sperm, providing direct evidence for the known 'slow block to polyspermy' provided by changes to the ZP adhesion/ability to be penetrated. Careful statistical analyses allow the authors to revisit the role of the ZP in preventing polyspermy: They show that the ZP block resulting from the cortical reaction is too slow (in the range of an hour) to contribute to the immediate prevention of polyspermy in mice. The presented analyses reveal that the ZP does contribute to the block to polyspermy in two other ways, namely by effectively limiting the number of sperm that reach the oocyte surface in a fertilization-independent manner, and by retaining components like JUNO and CD9, that are shed from the oocyte plasma membrane after fertilization, in the perivitelline space, which may help neutralize surplus spermatozoa that are already present in the PVS. Lastly, the authors report that the ZP may also contribute to channeling the flagellar oscillations of spermatozoa in the PVS to promote their fusion competence.

****Major comments:****

- Are the key conclusions convincing?

The authors provide a careful analysis of the dynamics of events, though the analyses are correlative, and can only be suggestive of causation. While this is a limitation of the study, it provides important analysis for future research. Moreover, by analysing also control oocytes without fertilization and the timing of events, the authors have in some instances clear 'negative controls' for comparison.

Some claims would benefit from rewording or rephrasing to put the findings better in the context of what is already known and what is novel:

- the phrasing 'challenging prior dogma' might be too strong since it had been observed before that it is not necessarily the first sperm that gets through the ZP that fertilizes the egg (though I am afraid that I do not have any citations or references for this). However, given that in the field people generally think it is not necessarily and always the first sperm, the authors may want to consider weakening this claim.

- I do think the cortical granule release could still contribute to the block to polyspermy though - as the authors here nicely show - at a later time-point only, and thus not the major and not the immediate block as previously thought. The wording in the abstract should therefore be adjusted (since it could still contribute...)

- the finding that the ZP presents a natural effective barrier for sperm entry is not that novel (as suggested here) - there are mutants that prevent sperm from getting through the ZP and thus to the oocyte and those lead to sterility

- release of OPM components - in the abstract it's unclear what the authors mean by this - in the results part it becomes clear. Please already make it clear in the abstract that it is the fertility factors JUNO/CD9 that could bind to sperm heads upon their release and thus 'neutralize' them? I would also recommend not referring to it as 'outer' plasma membrane (there is no 'inner plasma membrane'). Moreover, in the abstract please clarify that this release is happening only after fusion of the first sperm and not all the time. In the abstract it sounds as if this was a completely new idea, but there is good prior evidence that this is in fact happening (as also then cited in the results part) - maybe frame it more as the retention inside the PVS as new finding.

It is unclear to me what the relevance of dividing the post-fusion/post-engulfment into different phases as done in Fig 2 (phase 1, and phase 2) - also for the conclusions of this paper this seems rather irrelevant and overly complicated, since the authors never get back to it and don't need it (it's not related to the polyspermy block analyses). I would

remove it from the main figures and not divide into those phases since it is distracting from the main focus.

For the statistical analysis, I am not sure whether the assumption "assumption that the probability distribution of penetration or fertilization is uniform within a given time window" is in fact true since the probability of fertilizing decreases after the first fertilization event.... Maybe I misunderstood this, but this needs to be explained (or clarified) better, or the limitation of this assumption needs to be highlighted.

- Suggestion for additional experiments:

If I understood correctly, the onset of fusion in Fig 2C is defined by stopping of sperm beating? If it is by the sudden stop of the beating flagellum, this should be confirmed in this situation (with the ZP intact) that it correctly defines the time-point of fusion since this has not been measured in this set-up before as far as I understand. In order to measure this accurately, the authors will need to measure this accurately to be able to acquire those numbers (of time from fusion to end of engulfment), e.g. by pre-loading the oocyte with Hoechst to transfer Hoechst to the fusing sperm upon membrane fusion.

Fig 8: 2 comments

- To better show JUNO/CD9 pre-fusion attachment to the oocyte surface and post-fusion loss from the oocyte surface (but persistence in the PVS), an image after removal of the ZP (both for pre-fertilization and post-fertilization) would be helpful - the combination of those images with the ones you have (ZP intact) would make your point more visible.

- You show that the heads of spermatozoa post fusion are covered in CD9 and JUNO, yet I was missing an image of sperm in the PVS pre-fertilization (which should then not yet be covered).

****Minor comments:****

- The videos were remarkable to look at, and great to view in full. However, for the sake of time, the authors might want to consider cropping them for the individual phases to have a shorter video (with clear crop indicators) with the most important different stages visible in a for example 1 min video (e.g. video 1)

- In general, given that the ZP, PVS and oocyte membrane are important components, a general scheme at the very beginning outlining the relative positioning of each before and during fertilization (and then possibly also including the second polar body release) would be extremely helpful for the reader to orient themselves.

- first header results "Multi-penetration and polyspermy under in vivo conditions and standard and kinetics in vitro fertilization conditions" is hard to understand - simplify/make clearer (comparison of in vivo and in vitro conditions? Establishing the in vitro condition as assay?)

- Large parts of the statistical analysis (the more technical parts) could be moved to the methods part since it disrupts the flow of the text.

- To me, one of the main conclusions was given in the text of the results part, namely that "This suggests that first fertilization contributes effectively to the fertilization

- block, but less so to the penetration block". I would suggest that the authors use this conclusion to strengthen their rationale and storyline in the abstract.

- Wording: To characterize the kinetics with which penetration of spermatozoa in the PVS falls down after a first fertilization," falls down should be replaced with decreases (page 10 and page 12)

2. Significance:

Significance (Required)

Overall, this manuscript provides very interesting and carefully obtained data which provides important new insights particularly for reproductive biology. I applaud the authors on first establishing the in vivo conditions (how often do multiple sperm even penetrate the ZP in vivo) since studies have usually just started with in vitro condition where sperm at much higher concentration is added to isolated oocyte complexes. Thank you for providing an in vivo benchmark for the frequency of multiple sperm being in the PVS. While this frequency is rather low (somewhat expectedly, with 16% showing 2-3 sperm in the PVS), this condition clearly exists, providing a clear rationale for the investigation of mechanisms that can prevent additional sperm from entering.

My own expertise is experimentally - thus I don't have sufficient expertise to evaluate the statistical methods employed here.

3. How much time do you estimate the authors will need to complete the suggested revisions:

Estimated time to Complete Revisions (Required)

(Decision Recommendation)

Between 1 and 3 months

4. Review Commons values the work of reviewers and encourages them to get credit for their work. Select 'Yes' below to register your reviewing activity at Web of Science Reviewer Recognition Service (formerly Publons); note that the content of your review will not be visible on Web of Science.

No

Review #2

1. Evidence, reproducibility and clarity:

Evidence, reproducibility and clarity (Required)

Overall, this is a very interesting and relevant work for the field of fertilization. In general, the experimental strategies are adequate and well carried out. I have some questions and suggestions that should be considered before the work is published.

1. Why are the cumulus cells not mentioned when the AR is triggered before or while the sperms cross it? It seems the paper assumes from previous work that all sperm that reach ZP and the OPM have carried out the acrosome reaction. This, though probably correct, is still a matter of controversy and should be discussed. It is in a way strange that the authors do not make some controls using sperm from mice expressing GFP in the acrosome, as they have used in their previous work.

2. In the penetration block equations, it is not clear to me why ($tPF1$) refers to both $PIPF1$ and $\sigma\sigma PIPF1$. Is it as function off?

3. Why do the authors think that the flagella stops. The submission date was 2024-10-01 07:27:26 and there has been a paper in biorxiv for a while that merits mention and discussion in this work (bioRxiv [Preprint]. 2024 Jul 2:2023.06.22.546073. doi: 10.1101/2023.06.22.546073.PMID: 37904966).

4. Please correct at the beginning of Materials and Methods: Sperm was obtained from WT male mice, it should say were.

5. This is also the case in the fourth paragraph of this section: oocyte were not was.

2. Significance:

Significance (Required)

Understanding mammalian gamete fusion and polyspermy inhibition has not been fully achieved. The authors examined real time brightfield and confocal images of inseminated ZP-intact mouse oocytes and used statistical analyses to accurately determine the dynamics of the events that lead to fusion and involve polyspermy prevention under conditions as physiological as possible. Their kinetic observations in mice gamete interactions challenge present paradigms, as they document that the first sperm is not necessarily the one that fertilizes, suggesting the existence of other post-penetration fertilization factors. The authors find that the zona pellucida (ZP) block triggered by the cortical reaction is too slow to prevent polyspermy in this species. In contrast, their findings indicate that ZP directly contributes to the polyspermy block operating as a naturally effective entry barrier inhibiting the exit from the perivitelline space (PVS) of components released from the oocyte plasma membrane (OPM), neutralizing unwanted sperm fusion, aside from any block caused by fertilization. Furthermore, the authors unveil a new important ZP role regulating flagellar beat in fertilization by promoting sperm fusion in the PVS.

3. How much time do you estimate the authors will need to complete the suggested revisions:

Estimated time to Complete Revisions (Required)

(Decision Recommendation)

Less than 1 month

4. Review Commons values the work of reviewers and encourages them to get credit for their work. Select 'Yes' below to register your reviewing activity at Web of Science Reviewer Recognition Service (formerly Publons); note that the content of your review will not be visible on Web of Science.

No

Review #3

1. Evidence, reproducibility and clarity:

Evidence, reproducibility and clarity (Required)

****Summary:****

This study by Dubois et al. utilizes live-cell imaging studies of mouse oocytes undergoing fertilization. A strength of this study is their use of three different conditions for analyses of events of fertilization: (1) eggs undergoing fertilization retrieved from females at 15 hr after mating (n = 211 oocytes); (2) cumulus-oocyte complexes inseminated in vitro (n = 220 oocytes), and (3) zona pellucida (ZP)-intact eggs inseminated in vitro, transferred from insemination culture once sperm were observed bound to the ZP for subsequent live-cell imaging (93 oocytes). This dataset and these analyses are valuable for the field of fertilization biology. Limitations of this manuscript are challenges arise with some conclusions, and the presentation of the manuscript. There are some factual errors, and also some places where clearer explanations should to be provided, in the text and potentially augmented with illustrations to provide more clarity on the models that the authors interpret from their data.

****Major comments:****

The authors are congratulated on their impressive collection of data from live-cell imaging. However, the writing in several sections is challenging to understand or seems to be of questionable accuracy. The lack of accuracy is suspected to be more an effect of overly

ambitious attempts with writing style, rather than to mislead readers. Nevertheless, these aspects of the writing should be corrected. There also are multiple places where the manuscript contradicts itself. These contradictions should be corrected. Finally, there are factual points from previous studies that need correction.

Second, certain claims and the conclusions as presented are not always clearly supported by the data. This may be connected to the issues with writing style, word and phrasing choices, etc. The conclusions could be expressed more clearly, and thus may not require additional experiments or analyses to support them. The authors might also consider illustrations as ways to highlight the points they wish to make. (Figure 7 is a strong example of how they use illustrations to complement the text).

****Specific comments:****

1. The authors should use greater care in describing the blocks to polyspermy, particularly because they appear to be wishing to reframe views about prevention of polyspermic fertilization. The title mentions of "the fast block to polyspermy;" this problematic for a couple of different reasons. There is no strong evidence for block to polyspermy in mammals that occurs quickly, particularly not in the same time scale as the first-characterized fast block to polyspermy. To many biologists, the term "fast block to polyspermy" refers to the block that has been described in species like sea urchins and frogs, meaning a rapid depolarization of the egg plasma membrane. However, such depolarization events of the egg membrane have not been detected in multiple mammalian species. Moreover, the change in the egg membrane after fertilization does not occur in as fast a time scale as the membrane block in sea urchins and frogs (i.e., is not "fast" per se), and instead occurs in a comparable time frame as the conversation of the ZP associated with the cleavage of ZP2. Thus, it is misleading to use the terms "fast block" and "slow block" when talking about mammalian fertilization.

This also is an instance of where the authors contradict themselves in the manuscript, stating, "the membrane block and the ZP block are established in approximatively the same time frame" (third paragraph of Introduction). This statement is indeed accurate, unlike the reference to a fast block to polyspermy in mammals.

2. The authors aim to make the case that events occurring in the perivitelline space (PVS) prevent polyspermic fertilization, but the data that they present is not strong enough to make this conclusion. Additional experiments would be optional for this study, but data from such additional experiments are needed to support the authors' claims regarding these functions in fertilization. Without additional data, the authors need to be much more conservative in interpretations of their data. The authors have indeed observed phenomena (the presence of CD9 and JUNO in the PVS) that could be consistent with a molecular basis of a means to prevent fertilization by a second sperm. However, the authors would need additional data from additional experimental studies, such as interfering with the release of CD9 and JUNO and showing that this experimental manipulation leads to increased polyspermy, or creating an experimental situation that mimics the presence of CD9 and JUNO (in essence, what the authors call "sperm inhibiting medium" on page 20) and showing that this prevents fertilization.

A major section of the Results section here (starting with "The consequence is that ... ") is speculation. Rather than be in the Results section, this should be in the Discussion. The language should be also softened regarding the roles of these proteins in the perivitelline space in other portions of the manuscript, such as the abstract and the introduction.

Finally, the authors should do more to discuss their results with the results of Miyado et al. (2008), which interestingly, posited that CD9 is released from the oocytes and that this facilitates fertilization by rendering sperm more fusion-competent. There admittedly are two reports that present data that suggest lack of detection of CD9-containing exosomes from eggs (as proposed by Miyado et al.), but nevertheless, the authors should put their results in context with previous findings.

3. Many of the authors' conclusions focus on their prior analyses of sperm interaction - beautifully illustrated in Figure 7. However, the authors need to be cautious in their interpretations of these data and generalizing them to mammalian fertilization as a whole, because mouse and other rodent sperm have sperm head morphology that is quite different from most other mammalian species.

In a similar vein, the authors should be cautious in their interpretations regarding the extension of these results to mammalian species other than mouse, given data on numbers of perivitelline sperm (ranging from 100s in some species to virtually none in other species), suggesting that different species rely on different egg-based blocks to

polyspermy to varying extents. While these observations of embryos from natural matings are subject to numerous nuances, they nevertheless suggest that conclusions from mouse might not be able to be extended to all mammalian species.

4. Results, page 4 - It is very valuable that the authors clearly define what they mean by a penetrating spermatozoon and a fertilizing spermatozoon. However, they sometimes appear not to adhere to these definitions in other parts of the manuscript. An example of this is on page 10; the description of penetration of spermatozoon seems to be referring to membrane fusion with the oocyte plasma membrane, which the authors have alternatively called "fertilizing" or fertilization - although this is not entirely clear. The authors should go through all parts of the manuscript very carefully and ensure consistent use of their intended terminology.

Overall, while these definitions on page 4 are valuable, it is still recommended that the authors explicitly state when they are addressing penetration of the ZP and fertilization via fusion of the sperm with the oocyte plasma membrane. This help significantly in comprehension by readers. An example is the section header in the middle of page 9 - this could be "Spermatozoa can penetrate the ZP after the fertilization, but have very low chances to fertilize."

Another variation of this is in the middle of page 9, where the authors use the terms "fertilization block" and "penetration block." These are not conventional terms, and venture into being jargon, which could leave some readers confused. The authors could clearly define what they mean, particularly with respect to "penetration block,"

This extends to other portions of the manuscript as well, such as Figure 2C, with the label on the y-axis being "Time after fertilization." It seems that what the authors actually observed here was the cessation of sperm tail motility. (It is not evident they they did an assessment of sperm-oocyte fusion here.)

5. Several points that the authors try to make with several pieces of data do not come across clearly in the text, including Figure 2 on page 6, Figure 4 on page 9, and the various states utilized for the statistical treatment, "post-first penetration, post-first fertilization, no fertilization, penetration block and polyspermy block" on page 10 . Either re-writing and clearer definitions'explanations are needed, and/or schematic illustrations could be considered to augment re-written text. Illustrations could be a valuable way present the

intended concepts to readers more clearly and accurately. For example, Figure 4 and the associated text on page 9 get particularly confusing - although this sounds like a quite impressive dataset with observations of 138 sperm. Illustrations could be helpful, in the spirit of "a picture is worth 1000 words," to show what seem to be three different situations of sequences of events with the sperm they observed. Finally, the text in the Results about the 138 sperm is quite difficult to follow. It also might help comprehension to augment the percentages with the actual numbers of sperm - e.g., is 48.6% referring 67 of the total 138 sperm analyzed? Does the 85.1% refer to 57 of these 67 sperm?

6. Introduction, page 2 - it is inaccurate to state that only diploid zygotes can develop into a "new being." Triploid zygotes typically fail early in develop, but can survive and, for example, contribute to molar pregnancies. Additionally, it would be beneficial to be more scientifically precise term than saying "development into a new being." This is recommended not only for scientific accuracy, but also due to current debates, including in lay public circles, about what defines "life" or human life.

7. Introduction, page 2 - The mammalian sperm must pass through three layers, not just two as stated in the first paragraph of the Introduction. The authors should include the cumulus layer in this list of events of fertilization.

8. Introduction, page 2 - While there is evidence that zinc is released from mouse egg upon fertilization, the evidence is not convincing or conclusive that zinc is released from cortical granules or via cortical granule exocytosis.

9. The authors inaccurately state, "only if monospermic multi-penetrated oocytes are able to develop normally, which to our knowledge has never been proven in mice" (page 4) - This was demonstrated with the Astl knockout, assuming that the authors use of "multi-penetrated oocytes" here refers to the definition of penetration that they use, namely penetrating the ZP. This also is one of the instances where the authors contradict themselves, as they note the results with this knockout on page 18.

****Minor comments:****

There are numerous places where this reader marked places of confusion in the text. A sample of some of these:

Page 4 - "continuously relayed by other if they detach" - don't know what this means

Page 6 - "hernia" - do the authors mean "protrusion" on the oocyte surface?

Page 10 - "penetration of spermatozoa in the PVS falls down" - don't know what this means

Page 12 - "spermatozoa linked to the oocyte ZP" - not clear what "linked" means here

Page 14 - "by dint of oscillations" - don't know what this means

Specifics for Materials and Methods:

Exact timing of females receiving hCG and then being put with males for mating - assume this was immediate but this is an important detail regarding the timing for the creation of embryos in vivo.

Please provide the volumes in which inseminations occurred, and how many eggs were placed in this volume with the 10^6 sperm/ml.

****Referees cross-commenting****

I concur with Reviewer 1's comment, that the 'challenging prior dogma' about the first sperm not always being the one to fertilize the egg is too strong. As Reviewer 1 notes, "it had been observed before that it is not necessarily the first sperm that gets through the ZP that fertilizes the egg." I even thought about adding this comment to my review, although held off (I was hoping to find references, but that was taking too long).

2. Significance:

Significance (Required)

This manuscript brings interesting new observations for the field of gamete and fertilization biology. For very obvious reasons, the understanding of mammalian fertilization has lagged behind the understanding of fertilization of species with external fertilization. Decades ago, developmental biologists first focused on studies of fertilization on gametes from species that release sperm and egg into water, either spontaneously or with relatively easy stimulation, and gametes that could be easily cultured and enabled to create embryos as researchers watched. Studies of mammalian fertilization have since caught up, with the elucidation of conditions that support in vitro fertilization in various mammalian species, most notably mouse as an experimental model.

3. How much time do you estimate the authors will need to complete the suggested revisions:

Estimated time to Complete Revisions (Required)

(Decision Recommendation)

Between 1 and 3 months

4. Review Commons values the work of reviewers and encourages them to get credit for their work. Select 'Yes' below to register your reviewing activity at Web of Science Reviewer Recognition Service (formerly Publons); note that the content of your review will not be visible on Web of Science.

No

Full Revision

Manuscript number: RC-2024-02719

Corresponding author(s): Christine GOURIER and Nicolas RODRIGUEZ

[Please use this template only if the submitted manuscript should be considered by the affiliate journal as a full revision in response to the points raised by the reviewers.]

*If you wish to submit a preliminary revision with a revision plan, please use our "Revision Plan" template. **It is important to use the appropriate template to clearly inform the editors of your intentions.**]*

1. General Statements [optional]

This section is optional. Insert here any general statements you wish to make about the goal of the study or about the reviews.

All the three reviewers' comments on the original version of our manuscript have been fully addressed. Their input was extremely valuable in helping us clarify and refine the presentation of our results and conclusions. Their feedback contributed to making the study both more thoroughly developed and more accessible to a broad readership, while preserving its mechanistic depth. We believe that this revised version more effectively highlights the conceptual advances brought by our findings.

This section is mandatory. Please insert a point-by-point reply describing the revisions that were already carried out and included in the transferred manuscript.

Reviewer #1

Evidence, reproducibility and clarity

The manuscript "Key roles of the zona pellucida and perivitelline space in promoting gamete fusion and fast block to polyspermy inferred from the choreography of spermatozoa in mice oocytes" by Dr. Gourier and colleagues explores the poorly understood process of gamete fusion and the subsequent block to polyspermy by live-cell imaging of mouse oocytes with intact zona pellucida in vitro. The new component in this study is the presence of the ZP, which in prior studies of live-cell imaging had been removed before. This allowed the authors to examine contributions of the ZP to the block in polyspermy in relation to the timing of sperm penetrating the ZP and sperm fusing with the oocyte. By carefully analysing the timing of the cascade of events, the authors find that the first sperm that reaches the membrane of the mouse oocyte is not necessarily the one that fertilizes the oocytes, revealing that other mechanisms post-ZP-penetration influence the success of individual sperm. While the rate of ZP penetration remains constant in unfertilized oocytes, it decreases upon fertilization for subsequent sperm, providing direct evidence for the known 'slow block to polyspermy' provided by changes to the ZP adhesion/ability to be penetrated. Careful statistical analyses allow the authors to revisit the role of the ZP in preventing

Full Revision

polyspermy: They show that the ZP block resulting from the cortical reaction is too slow (in the range of an hour) to contribute to the immediate prevention of polyspermy in mice. The presented analyses reveal that the ZP does contribute to the block to polyspermy in two other ways, namely by effectively limiting the number of sperm that reach the oocyte surface in a fertilization-independent manner, and by retaining components like JUNO and CD9, that are shed from the oocyte plasma membrane after fertilization, in the perivitelline space, which may help neutralize surplus spermatozoa that are already present in the PVS. Lastly, the authors report that the ZP may also contribute to channeling the flagellar oscillations of spermatozoa in the PVS to promote their fusion competence.

Major comments:

- Are the key conclusions convincing?

The authors provide a careful analysis of the dynamics of events, though the analyses are correlative, and can only be suggestive of causation. While this is a limitation of the study, it provides important analysis for future research. Moreover, by analysing also control oocytes without fertilization and the timing of events, the authors have in some instances clear 'negative controls' for comparison.

Some claims would benefit from rewording or rephrasing to put the findings better in the context of what is already known and what is novel:

- the phrasing 'challenging prior dogma' might be too strong since it had been observed before that it is not necessarily the first sperm that gets through the ZP that fertilizes the egg (though I am afraid that I do not have any citations or references for this). However, given that in the field people generally think it is not necessarily and always the first sperm, the authors may want to consider weakening this claim.

Only real-time imaging of in vitro fertilization of zona pellucida-intact oocytes, as performed in our study, is capable of determining which spermatozoon crossing the zona pellucida fuses with the oocyte. However, such studies are rare, and most do not specifically address this question. As Reviewers 1 & 3, we have not found any citation or reference telling or showing that it is not necessarily the first spermatozoon to penetrate the zona pellucida that fertilizes the egg. In contrast, at least one reference (Sato et al., 1979) explicitly reports the opposite. If, as suggested by Reviewer 1 and 3, it has indeed been observed before that the first sperm to pass the ZP is not always the one that fertilizes, and if this idea is generally accepted in the field, then it is all the more important that a study demonstrates and publishes this point. This is precisely what our study makes possible. However, in case we may have overlooked a previous reference making the same observation as ours, we have removed the phrasing 'challenging prior dogma'. That being said, the key issue is not so much that it is not necessarily the first spermatozoon

Full Revision

penetrating the perivitelline space that fertilizes, but rather why spermatozoa that successfully reach the PVS of an unfertilized oocyte may fail to achieve fertilization. This is one of the central questions our study sought to address.

- I do think the cortical granule release could still contribute to the block to polyspermy though - as the authors here nicely show - at a later time-point only, and thus not the major and not the immediate block as previously thought. The wording in the abstract should therefore be adjusted (since it could still contribute...)

We are concerned that we may disagree on this point. The penetration block resulting from cortical granule release progressively reduces the permeability of the zona pellucida to spermatozoa, relative to its baseline permeability prior to sperm-oocyte fusion. Any decrease in this baseline permeability occurring before the fusion block becomes fully effective can contribute to the prevention of polyspermy by limiting the number of sperm that can access the oolemma at a time when fusion is still possible. In contrast, once the fusion block is fully established, limiting the number of spermatozoa traversing the ZP becomes irrelevant regarding the block to polyspermy, as the fusion block alone is sufficient to prevent additional fertilizations, rendering the penetration block obsolete. The only scenario that could challenge this obsolescence is if the fusion block were transient. In that case, as Reviewer 1 suggests, the penetration block could indeed play a role at a later time-point. However, taken together, our study and that of Nozawa et al. (2018) support the conclusion that this is not the case in mice:

- (i) Our in vitro study using kinetic tracking shows that the time constant for completion of the fusion block is typically 6.2 ± 1.3 minutes. During this time window, we observe that the permeability of the zona pellucida to spermatozoa does not yet decrease significantly from the baseline level it exhibited prior to sperm-oocyte fusion (see Figures 5B and S1B in the revised manuscript, and Figures 5A and 5B in the initial version). Consequently, before the fusion block is fully established, the penetration block can contribute only marginally—if at all—to the prevention of polyspermy. In contrast, the naturally low baseline permeability of the ZP—independent of any fertilization-triggered penetration block—as well as the relatively long timing of fusion (15.8 ± 5.7 minutes on average) after sperm penetration in the perivitelline space, are factors that contribute to the preservation of monospermic while the fusion block is still being established.
- (ii) Our in vitro study using kinetic tracking shows that once the fusion block is completed following the first fusion event, no additional spermatozoa are able to fuse with the oocyte until the end of the experiment, 4 hours post-insemination (see blue points and fitting curve in Figure 5C). Meanwhile, one or more additional spermatozoa—most of them motile and therefore viable—are present in the perivitelline space in 50% of the oocytes analyzed (purple point in Figure 5C). This demonstrates that, once established, the fusion block remains effective for at least the entire duration of the experiment, supporting the idea of a fully functional and long-lasting fusion block.
- (iii) Nozawa et al. (2018) found that female mice lacking ovastacin—the protease released during the cortical reaction that renders the zona pellucida impenetrable—are normally

Full Revision

fertile. They additionally reported that the oocytes recovered from these females after mating are monospermic despite the systematic presence of additional spermatozoa in the perivitelline space. These findings further support the conclusion that in mice the fusion block is both permanent and sufficient to prevent polyspermy.

For all these reasons, we believe that even at a later time-point, the penetration block does not contribute to the prevention of polyspermy in mice.

To clarify the fact that the penetration block does not necessarily contribute to prevent polyspermy, which indeed challenges the commonly accepted view, we have substantially revised the discussion. Furthermore, Figure 9 from the initial version of the manuscript has been replaced by Figure 8 in the revised version. This new figure provides a more didactic illustration of the inefficacy of the penetration block in preventing polyspermy in mice, by showing the respective impact of the fusion block, the penetration block, as well as fusion timing and the natural baseline permeability of the zona pellucida, on the occurrence of polyspermy.

As for the abstract, it has also been thoroughly revised. The content related to this section is now expressed in a way that emphasizes the factors that actively contribute to the prevention of polyspermy in mice, rather than those with no or marginal contribution (such as the penetration block in this case).

- release of OPM components - in the abstract it's unclear what the authors mean by this - in the results part it becomes clear. Please already make it clear in the abstract that it is the fertility factors JUNO/CD9 that could bind to sperm heads upon their release and thus 'neutralize' them? I would also recommend not referring to it as 'outer' plasma membrane (there is no 'inner plasma membrane'). Moreover, in the abstract please clarify that this release is happening only after fusion of the first sperm and not all the time. In the abstract it sounds as if this was a completely new idea, but there is good prior evidence that this is in fact happening (as also then cited in the results part) - maybe frame it more as the retention inside the PVS as new finding.

We thank reviewer 1 for pointing out the lack of precision in the abstract regarding the "components" released from the oolemma, and the fact that our phrasing may have given the impression that the post-fertilization release of CD9 and JUNO is a novel observation. The new observation is that CD9 and JUNO, which are known to be massively released from the oolemma after fertilization, bind to spermatozoa in the perivitelline space. However, we cannot rule out the possibility that other oocyte-derived molecules not investigated here may undergo a similar process. This is why we employed the broader term "components", which encompasses both CD9 and JUNO as well as potential additional molecules. That said, we acknowledge the lack of precision introduced by this terminology. To address this, we have revised the corresponding sentence in the abstract to better reflect our new findings relative to previous ones, and to eliminate the ambiguity introduced by the word "component".

The revised sentence of the abstract reads as follows:

Full Revision

“Our observation that non-fertilizing spermatozoa in the perivitelline space are coated with CD9 and JUNO oocyte’s proteins, which are known to be massively released from the oolemma after gamete fusion, supports the hypothesis that the fusion block involves an effective perivitelline space-block contribution consisting in the neutralization of supernumerary spermatozoa in the perivitelline space by these and potentially other oocyte-derived factors.”

Moreover, we cannot state in the abstract that the release of CD9 and JUNO occurs only after the fusion of the first spermatozoon and not before, since some CD9 and JUNO are already detectable in the perivitelline space (PVS) prior to fusion. What our study shows is that, before fertilization, CD9 and JUNO are predominantly localized at the oocyte membrane. In contrast, after fusion (four hours post-insemination), oocyte CD9 is distributed between the membrane and the PVS, and the only JUNO signal detectable in the oocyte is found in the PVS. This is what we describe in the Results section on page 15.

Regarding the acronym “OPM” in the initial version of the manuscript, although it was defined in the introduction as referring to the **oocyte plasma membrane** and not the **outer plasma membrane** (which, indeed, would not be meaningful), we acknowledge that it may have caused confusion to people in the field due to its resemblance to the commonly used meaningful acronym “OAM” for **outer acrosomal membrane**. To avoid any ambiguity, we have replaced the acronym “OPM” throughout the revised manuscript with the term “oolemma”, which unambiguously refers to the plasma membrane of the oocyte.

It is unclear to me what the relevance of dividing the post-fusion/post-engulfment into different phases as done in Fig 2 (phase 1, and phase 2) - also for the conclusions of this paper this seems rather irrelevant and overly complicated, since the authors never get back to it and don't need it (it's not related to the polyspermy block analyses). I would remove it from the main figures and not divide into those phases since it is distracting from the main focus.

Sperm engulfment and PB2 extrusion are two processes that follow sperm–oocyte fusion. As such, they are clear indicators that fusion has occurred and that meiosis has resumed. Their progression over time is readily identifiable in bright-field imaging: sperm engulfment is characterized by the gradual disappearance of the spermatozoon head from the oolemma, whereas PB2 extrusion is observed as the progressive emergence of a rounded protrusion from the oocyte membrane (Figure 2 in the initial manuscript and Figure S2 A&B in the revised version). The kinetics of these events, measured from the arrest of “push-up–like” movement of the sperm head against the oolemma —assumed to coincide with sperm–oocyte fusion, as further justified in a later response to Reviewer 1—provide reliable temporal landmarks for estimating the timing of fusion when the fusion event itself is not directly observed in real time (Figure S2 C&D).

The four landmarks used in this estimation are:

- (i) the disappearance of the sperm head from the oolemma due to internalization (28 ± 2 minutes post-arrest, mean \pm SD);
- (ii) the onset of PB2 protrusion from the oolemma (28 ± 2 minutes post-arrest);

Full Revision

(iii) the moment when the contact angle between the PB2 protrusion and the oolemma shifts from greater than to less than 90° (49 ± 6 minutes post-arrest);

(iv) the completion of PB2 extrusion (73 ± 10 minutes post-arrest).

The approach used to determine the fusion time window of a fertilizing spermatozoon from these landmarks is detailed in the “*Determination of the Fertilization Time Windows*” section of the Materials and Methods. Compared to the initial version of the manuscript, we have added a paragraph explaining the rationale for using the arrest of the push-up-like movement as a reliable indicator for sperm–oocyte fusion and have clarified the description of the approach used to determine fertilization timing.

The timed characterization of sperm engulfment and PB2 extrusion kinetics is highly relevant to the analysis of the penetration and fusion blocks, however we agree that its place is more appropriate in the Supplementary Information than in the main text. In accordance with the reviewer’s recommendation, this section has therefore been moved to the Supplementary Information SI2.

For the statistical analysis, I am not sure whether the assumption “assumption that the probability distribution of penetration or fertilization is uniform within a given time window” is in fact true since the probability of fertilizing decreases after the first fertilization event.... Maybe I misunderstood this, but this needs to be explained (or clarified) better, or the limitation of this assumption needs to be highlighted.

During in vitro fertilization experiments with kinetic tracking, each oocyte is observed sequentially in turn. As a result, sperm penetration into the perivitelline space or fusion with the oolemma may occur either during an observation round or in the interval between two rounds. In the former case, penetration or fusion is directly observed in real time, allowing for high temporal precision in determining the moment of the event. In contrast, when penetration or fusion occurs between two observation rounds, the precise timing cannot be directly determined. We can only ascertain that the event took place within the time window we have determined. Because, within a given penetration or fusion time window, we do not know the exact moment at which the event occurred, there is no reason to favor one time over another. This justifies the assumption that all time points within the window are equally probable. This explanation has been added in the section *Statistical treatment of penetration and fertilization chronograms to study the kinetics of fertilization, penetration block and fusion block* of the main text and in the section *Statistical treatment of penetrations and fertilizations chronograms to study penetration and fusion blocks* of the material and methods.

-Suggestion for additional experiments:

If I understood correctly, the onset of fusion in Fig 2C is defined by stopping of sperm beating? If it is by the sudden stop of the beating flagellum, this should be confirmed in this situation (with the ZP intact) that it correctly defines the time-point of fusion since this has not been measured in this set-up before as far as I understand. In order to measure this accurately, the authors will need

Full Revision

to measure this accurately to be able to acquire those numbers (of time from fusion to end of engulfment), e.g. by pre-loading the oocyte with Hoechst to transfer Hoechst to the fusing sperm upon membrane fusion.

The nuclear dye Hoechst is widely used as a marker of gamete fusion, as it transfers from the ooplasm—when preloaded with the dye—into the sperm nucleus upon membrane fusion, thereby signaling the happening of the fusion event. This technique is applicable in the context of in vitro fertilization using ZP-free oocytes. However, it is not suitable when cumulus–oocyte complexes are inseminated, as is the case in both in vitro experimental conditions of the present study (standard IVF and IVF with kinetic tracking). Indeed, when cumulus–oocyte complexes are incubated with Hoechst to preload the oocytes, the numerous surrounding cumulus cells also take up the dye. Consequently, upon insemination, spermatozoa acquire fluorescence while traversing and dispersing the cumulus mass—before reaching the ZP—thus rendering Hoechst labeling ineffective as a specific marker of membrane fusion. This remains true even under optimized conditions involving brief Hoechst incubation of cumulus–oocyte complexes (<5 minutes) and multiple subsequent washes (four incubations of 15 minutes each in fresh medium) prior to insemination.

Nonetheless, we have strong evidence supporting the use of the arrest of sperm movement as a surrogate marker for the moment of fusion. In our previous study (Ravaux et al., 2016; ref. 4 in the revised manuscript), we investigated the temporal relationship between the abrupt cessation of sperm head movement on the oolemma—resulting from strong flagellar beating arrest—and the fusion event, using ZP-free oocytes preloaded with Hoechst. That study revealed a temporal delay of less than one minute between the cessation of sperm oscillations and the actual membrane fusion, thereby supporting the conclusion that in ZP-free oocytes, the arrest of vigorous sperm movement at the oolemma is a reliable indicator of the moment at which fusion occurs. In the same study, the kinetics of sperm head internalization into the ooplasm were also characterized, typically concluding within 20–30 minutes after movement cessation. These findings are fully consistent with our current observations in ZP-intact oocytes, where sperm head engulfment was completed approximately 24 ± 3 minutes after the arrest of sperm oscillations. Taken together, these results strongly support the conclusion that, in both ZP-free and ZP-intact oocytes, the arrest of sperm movement is a reliable indicator of the fusion event. This assumption formed the basis for our determination of fertilization time points in the present study.

These justifications were not fully detailed in the original version of the manuscript. We have addressed this in the revised version by explicitly presenting this rationale in the Materials and Methods section under *Determination of the Fertilization Time Windows*.

Fig 8: 2 comments

- To better show JUNO/CD9 pre-fusion attachment to the oocyte surface and post-fusion loss from the oocyte surface (but persistence in the PVS), an image after removal of the ZP (both for

Full Revision

pre-fertilization and post-fertilization) would be helpful - the combination of those images with the ones you have (ZP intact) would make your point more visible.

We have followed this recommendation. Figure 8 of the initial manuscript has been replaced by Figure 6 in the revised manuscript, which illustrates the four situations encountered in this study: fertilized and unfertilized oocytes, each with and without unfused spermatozoa in their PVS. To better show JUNO/CD9 pre-fusion presence to the oocyte plasma membrane, as well as their post-fusion partial (for CD9) and near-complete (for JUNO) loss from the oocyte membrane (but persistence in the PVS), paired images of the same oocyte before and after of ZP removal are now provided, both for unfertilized (Figure 6A) and fertilized oocytes (Figure 6C).

- You show that the heads of spermatozoa post fusion are covered in CD9 and JUNO, yet I was missing an image of sperm in the PVS pre-fertilization (which should then not yet be covered).

As staining and confocal imaging of the oocytes were performed 4 hours after insemination, images of sperm in the PVS of an oocyte “pre-fertilization” cannot be strictly obtained. However, we can have images of spermatozoa present in the PVS of oocytes that remained unfertilized. This situation, now illustrated in Figure 6B of the revised manuscript, shows that these spermatozoa are also covered in JUNO and CD9, which they may have progressively acquired over time from the baseline presence of these proteins in the PVS of unfertilized oocytes. This also may provide a mechanistic explanation for their inability to fuse with the oolemma, and, consequently, for the failure of fertilization in these oocytes.

Minor comments:

- The videos were remarkable to look at, and great to view in full. However, for the sake of time, the authors might want to consider cropping them for the individual phases to have a shorter video (with clear crop indicators) with the most important different stages visible in a for example 1 min video (e.g. video).

We have followed this recommendation. The videos have been cropped and annotated in order to highlight the key events that support the points made in the result section from page 9 to 11 in the revised manuscript.

- In general, given that the ZP, PVS and oocyte membrane are important components, a general scheme at the very beginning outlining the relative positioning of each before and during fertilization (and then possibly also including the second polar body release) would be extremely helpful for the reader to orient themselves.

A general scheme addressing Reviewer 1 request, summarizing the key components and concepts discussed in the article and intended to help guide the reader, has been added to the introduction of the revised manuscript as Figure 1.

Full Revision

- first header results "Multi-penetration and polyspermy under in vivo conditions and standard and kinetics in vitro fertilization conditions" is hard to understand - simplify/make clearer (comparison of in vivo and in vitro conditions? Establishing the in vitro condition as assay?)

The title of the first Results section has been revised in accordance with Reviewer 1 suggestion. It now reads: *Comparative study of penetration and fertilization rates under in vivo and two distinct in vitro fertilization conditions.*

- Large parts of the statistical analysis (the more technical parts) could be moved to the methods part since it disrupts the flow of the text.

In the revised version of our manuscript, we have restructured this part of the analysis to ensure that more technical or secondary elements do not disrupt the flow of the main text. Accordingly, the equations have been reduced to only what is strictly necessary to understand our approach, their notation has been greatly simplified, and the statistical analysis of unfertilized oocytes whose zona pellucida was traversed by one or more spermatozoa has been moved to the Supplementary Information (SI1).

- To me, one of the main conclusions was given in the text of the results part, namely that "This suggests that first fertilization contributes effectively to the fertilization-block, but less so to the penetration block". I would suggest that the authors use this conclusion to strengthen their rationale and storyline in the abstract.

We agree with Reviewer 1 suggestion. Accordingly, we have not only thoroughly revised our abstract, but also the introduction and discussion, in order to better highlight the rationale of our study, its storyline, and the new findings which not only challenge certain established views but also open new research directions in the mechanisms of gamete fusion and polyspermy prevention.

- Wording: To characterize the kinetics with which penetration of spermatozoa in the PVS falls down after a first fertilization," falls down should be replaced with decreases (page 10 and page 12)

Falls down has been removed from the new version and replaced with decreases

Significance

Overall, this manuscript provides very interesting and carefully obtained data which provides important new insights particularly for reproductive biology. I applaud the authors on first establishing the in vivo conditions (how often do multiple sperm even penetrate the ZP in vivo) since studies have usually just started with in vitro condition where sperm at much higher concentration is added to isolated oocyte complexes. Thank you for providing an in vivo benchmark for the frequency of multiple sperm being in the PVS. While this frequency is rather

Full Revision

low (somewhat expectedly, with 16% showing 2-3 sperm in the PVS), this condition clearly exists, providing a clear rationale for the investigation of mechanisms that can prevent additional sperm from entering.

My own expertise is experimentally - thus I don't have sufficient expertise to evaluate the statistical methods employed here.

Full Revision

Reviewer #2

Evidence, reproducibility and clarity

Overall, this is a very interesting and relevant work for the field of fertilization. In general, the experimental strategies are adequate and well carried out. I have some questions and suggestions that should be considered before the work is published.

1) Why are the cumulus cells not mentioned when the AR is triggered before or while the sperms cross it? It seems the paper assumes from previous work that all sperm that reach ZP and the OPM have carried out the acrosome reaction. This, though probably correct, is still a matter of controversy and should be discussed. It is in a way strange that the authors do not make some controls using sperm from mice expressing GFP in the acrosome, as they have used in their previous work.

We do not mention the cumulus cells or whether the acrosome reaction is triggered before, during, or after their traversal (i.e., upon sperm binding to the ZP), as this question, while scientifically relevant, pertains to a distinct line of investigation that lies beyond the scope of the present study. Even with the use of spermatozoa expressing GFP in the acrosome, addressing this question would require a complete redesign of our kinetic tracking protocol, which was specifically conceived to monitor in bright field the dynamic behavior of spermatozoa from the moment they begin to penetrate the perivitelline space of an oocyte. Accordingly, we imaged oocytes that were isolated 15 minutes after insemination of the cumulus–oocyte complexes, by which time most (if not all) cumulus cells had detached from the oocytes, as explained in the fourth paragraph of the material and methods of both the initial and revised versions of the manuscript. The spermatozoa we had access to were therefore already bound to the zona pellucida at the time of removal from the insemination medium, and had thus necessarily passed through the cumulus layer. It is unclear for us why Reviewer 2 believes that we “assume from previous work that all sperm that reach ZP has carried out the acrosome reaction”. We could not find any statement in our manuscript suggesting, let alone asserting, such an assumption, which we know to be incorrect. Based on both published work from Hirohashi’s group in 2011 (Jin et al., 2011, DOI: 10.1073/pnas.1018202108) and our own unpublished observation (both involving cumulus-oocyte masses inseminated with spermatozoa expressing GFP in the acrosome), it is established that only a subset of spermatozoa reaching the ZP after crossing the cumulus layer has undergone acrosome reaction. Moreover, from the same sources—as well as from a recent publication by Buffone’s group (Jabloňsky et al., 2023 DOI: 10.7554/eLife.93792) which is the one to which reviewer 2 refers in her/his 3rd comment, it is also well established that spermatozoa have all undergone acrosome reaction when they enter the PVS. To the best of our knowledge, this latter point has long been widely accepted and is not questioned. Therefore, stating this in the first paragraph of the Discussion in the revised manuscript, while referencing the two aforementioned published studies, should be appropriate. What remains a matter of ongoing debate, however, is the timing and the physiological trigger(s) of the acrosome reaction in fertilizing spermatozoa. The 2011 study by Hirohashi’s group challenged the previously accepted view that ZP binding induces the acrosome reaction, showing instead that most spermatozoa capable of crossing the ZP and fertilizing the oocyte had already undergone the acrosome

Full Revision

reaction prior to ZP binding. However, as this issue lies beyond the scope of our study, we do not consider it appropriate to include a discussion of it in the manuscript.

2) In the penetration block equations, it is not clear to me why (t_{PF1}) refers to both PIPF1 and σ_{PIPF1} . Is it as function off?

That is correct: (t_{PF1}) means function of the time post-first fertilization. Both the post-first fertilization penetration index (i.e. PI_{PF1}) and its uncertainty (i.e. σ_{PIPF1}) vary as a function of this time. However, as mentioned in a previous response to Reviewer 1, this section has been rewritten to improve clarity and readability. The equations have been limited to those strictly necessary for understanding our approach, and their notation has been significantly simplified.

3) Why do the authors think that the flagella stops. The submission date was 2024-10-01 07:27:26 and there has been a paper in biorxiv for a while that merits mention and discussion in this work (bioRxiv [Preprint]. 2024 Jul 2:2023.06.22.546073. doi: 10.1101/2023.06.22.546073.PMID: 37904966).

Our experimental approach allows us to determine when the spermatozoon stops moving, but not why it stops. We thank Reviewer 3 for pointing out this very relevant paper from Buffone's group (doi: 10.7554/eLife.93792) which shows the existence of two distinct populations of live, acrosome-reacted spermatozoa. These correspond to two successive stages, which occur either immediately upon acrosome reaction in a subset of spermatozoa, or after a variable delay in others, during which the sperm transitions from a motile to an immotile state. The transition from the first to the second stage was shown to follow a defined sequence: an increase in the sperm calcium concentration, followed by midpiece contraction associated with a local reorganization of the helical actin cortex, and ultimately the arrest of sperm motility. For fertilizing spermatozoa in the PVS, this transition was shown to occur upon fusion. However, it was also reported in some non-fertilizing spermatozoa that this transition took place within the PVS. These findings are consistent with the requirement for sperm motility in order to achieve fusion with the oolemma. Moreover, the fact that some spermatozoa may prematurely transition to the immotile state within the PVS can therefore be added to the list of possible reasons why a spermatozoon that penetrates the PVS of an oocyte might fail to fuse.

This discussion has been added to the first paragraph of the Discussion section of our revised manuscript.

4) Please correct at the beginning of Materials and Methods: Sperm was obtained from WT male mice, it should say were.

Thank you, the correction has been done.

Full Revision

5) This is also the case in the fourth paragraph of this section: oocyte were not was.

The sentence in question has been modified as followed: “In the in vitro fertilization experiments with kinetic tracking, a subset of oocytes—together with their associated ZP-bound spermatozoa—was isolated 15 minutes post-insemination and transferred individually into microdrops of fertilization medium to enable identification.”

Significance

Understanding mammalian gamete fusion and polyspermy inhibition has not been fully achieved. The authors examined real time brightfield and confocal images of inseminated ZP-intact mouse oocytes and used statistical analyses to accurately determine the dynamics of the events that lead to fusion and involve polyspermy prevention under conditions as physiological as possible. Their kinetic observations in mice gamete interactions challenge present paradigms, as they document that the first sperm is not necessarily the one that fertilizes, suggesting the existence of other post-penetration fertilization factors. The authors find that the zona pellucida (ZP) block triggered by the cortical reaction is too slow to prevent polyspermy in this species. In contrast, their findings indicate that ZP directly contributes to the polyspermy block operating as a naturally effective entry barrier inhibiting the exit from the perivitelline space (PVS) of components released from the oocyte plasma membrane (OPM), neutralizing unwanted sperm fusion, aside from any block caused by fertilization. Furthermore, the authors unveil a new important ZP role regulating flagellar beat in fertilization by promoting sperm fusion in the PVS.

Reviewer #3 (Evidence, reproducibility and clarity (Required)):

SUMMARY:

This study by Dubois et al. utilizes live-cell imaging studies of mouse oocytes undergoing fertilization. A strength of this study is their use of three different conditions for analyses of events of fertilization: (1) eggs undergoing fertilization retrieved from females at 15 hr after mating (n = 211 oocytes); (2) cumulus-oocyte complexes inseminated in vitro (n = 220 oocytes), and (3) zona pellucida (ZP)-intact eggs inseminated in vitro, transferred from insemination culture once sperm were observed bound to the ZP for subsequent live-cell imaging (93 oocytes). This dataset and these analyses are valuable for the field of fertilization biology. Limitations of this manuscript are challenges arise with some conclusions, and the presentation of the manuscript. There are some factual errors, and also some places where clearer explanations should to be provided, in the text and potentially augmented with illustrations to provide more clarity on the models that the authors interpret from their data.

MAJOR COMMENTS:

The authors are congratulated on their impressive collection of data from live-cell imaging.

Full Revision

However, the writing in several sections is challenging to understand or seems to be of questionable accuracy. The lack of accuracy is suspected to be more an effect of overly ambitious attempts with writing style, rather than to mislead readers. Nevertheless, these aspects of the writing should be corrected. There also are multiple places where the manuscript contradicts itself. These contradictions should be corrected. Finally, there are factual points from previous studies that need correction.

Second, certain claims and the conclusions as presented are not always clearly supported by the data. This may be connected to the issues with writing style, word and phrasing choices, etc. The conclusions could be expressed more clearly, and thus may not require additional experiments or analyses to support them. The authors might also consider illustrations as ways to highlight the points they wish to make. (Figure 7 is a strong example of how they use illustrations to complement the text).

In response to Reviewer 3's concern about the writing style, which made several sections difficult to understand, we have thoroughly revised the entire manuscript to improve clarity, and precision. To further enhance comprehension, we have added illustrations in the revised version of the manuscript:

- **Figure 1A** presents the gamete components; **Figure 1B** depicts the main steps of fertilization considered in the present study; and **Figure 1C** illustrates the penetration and fusion blocks, along with the respective contributing mechanisms: the ZP-block for the penetration block, and the membrane-block and PVS-block for the fusion block

- **Figure 2A** provides a description of the three experimental protocols used in this study: Condition 1, in vivo fertilization after mating; Condition 2, standard in vitro fertilization following insemination of cumulus-oocyte complexes; and Condition 3, in vitro fertilization with kinetic tracking of oocytes isolated from the insemination medium 15 min after insemination of the cumulus-oocyte complexes.

- **Figure 4** (formerly Figure 7 in the initial version) now highlights all fusing and non-fusing situations documented in videos 1-6 and associated paragraphs of the Results section.

- In the **Discussion**, **Figure 9** from the original version has been replaced by **Figure 8**, which now provides a more pedagogical illustration of the inefficacy of the penetration block in preventing polyspermy in mice. This figure illustrates the respective contributions of the fusion block, the penetration block, fusion timing, and the intrinsic permeability of the zona pellucida to the occurrence of polyspermy.

We hope that this revised version of the article will guide the reader smoothly throughout, without causing confusion.

Full Revision

Regarding the various points that Reviewer 3 perceives as contradictions or factual errors, or the claims and the conclusions which, as presented, should not always supported by the data, we will provide our perspective on each of them as they are raised in the review.

SPECIFIC COMMENTS:

(1) The authors should use greater care in describing the blocks to polyspermy, particularly because they appear to be wishing to reframe views about prevention of polyspermic fertilization. The title mentions of "the fast block to polyspermy;" this problematic for a couple of different reasons. There is no strong evidence for block to polyspermy in mammals that occurs quickly, particularly not in the same time scale as the first-characterized fast block to polyspermy. To many biologists, the term "fast block to polyspermy" refers to the block that has been described in species like sea urchins and frogs, meaning a rapid depolarization of the egg plasma membrane. However, such depolarization events of the egg membrane have not been detected in multiple mammalian species. Moreover, the change in the egg membrane after fertilization does not occur in as fast a time scale as the membrane block in sea urchins and frogs (i.e., is not "fast" per se), and instead occurs in a comparable time frame as the conversation of the ZP associated with the cleavage of ZP2. Thus, it is misleading to use the terms "fast block" and "slow block" when talking about mammalian fertilization. This also is an instance of where the authors contradict themselves in the manuscript, stating, "the membrane block and the ZP block are established in approximatively the same time frame" (third paragraph of Introduction). This statement is indeed accurate, unlike the reference to a fast block to polyspermy in mammals.

We fully agree with Reviewer 3 on the importance of clearly defining the two blocks examined in the present study—the penetration block and the fusion block (as referred to in the revised version)—and of situating them in relation to the three blocks described in the literature: the ZP-block, membrane-block, and PVS-block. We acknowledge that this distinction was not sufficiently clear in the original version of the manuscript. In the revised version, these two blocks and their relationship to the ZP-, membrane-, and PVS-blocks are now clearly introduced in the second paragraph of the Introduction section and illustrated in the first figure of the manuscript (Fig. 1C). They are then discussed in detail in two dedicated paragraphs of the Discussion, entitled Relation between the penetration block and the ZP-block and Relation between the fusion block and the membrane- and PVS-blocks.

The *penetration block* refers to the time-dependent decrease in the number of spermatozoa penetrating the perivitelline space (PVS) following fertilization, whereas the *fusion block* refers to the time-dependent decrease in sperm-oolemma fusion events after fertilization. It is precisely to the characterization of these two blocks that our in vitro fertilization experiments with kinetic tracking allow us to access.

In this study, as in the literature, fusion-triggered modifications of the ZP that hinder sperm traversal of the ZP are referred to as the ZP-block (also known as ZP hardening). The ZP-block thus contributes to the post-fertilization reduction in sperm penetration into the PVS and thereby

Full Revision

underlies the penetration block. Similarly, fusion-triggered alterations of the PVS and the oolemma that reduce the likelihood of spermatozoa that have reached the PVS successfully to fuse with the oolemma are referred to as the PVS-block and membrane-block, respectively. These two blocks act together to reduce the probability of sperm-oolemma fusion after fertilization, and thus contribute to the fusion block.

The time constant of the penetration block was found to be 48.3 ± 9.7 minutes, which is consistent with the typical timeframe of ZP-block completion—approximately one hour post-fertilization in mice—as reported in the literature. By contrast, the time constant of the fusion block was determined to be 6.2 ± 1.3 minutes, which is markedly faster than the time typically reported in the literature for the completion of the fusion-block (more than one hour in mice). This strongly suggests that the kinetics of the fusion block are not primarily governed by its membrane-block component, but rather by its PVS-block component—about which little to nothing was previously known.

Contrary to what Reviewer 3 appears to have understood from our initial formulation, there is therefore no contradiction or error in stating that "the membrane block and the ZP block are established within approximately the same timeframe", while the fusion block, which proceeds much more rapidly, is likely to rely predominantly on the PVS-block. We have thoroughly revised the manuscript to clarify this key message of the study.

However, we understand Reviewer 3's objection to referring to the fusion block (or the PVS-block) as a fast block, given that this term is conventionally reserved for the immediate fertilization-triggered membrane depolarization occurring in sea urchins and frogs. Although the kinetics we report for the fusion block are considerably faster than those of the penetration block, they occur on the scale of minutes, and not seconds. In line with the reviewer's recommendation, we have therefore modified both the title and the relevant passages in the text to remove all references to the term fast block in the revised version.

(2) The authors aim to make the case that events occurring in the perivitelline space (PVS) prevent polyspermic fertilization, but the data that they present is not strong enough to make this conclusion. Additional experiments would be optional for this study, but data from such additional experiments are needed to support the authors' claims regarding these functions in fertilization. Without additional data, the authors need to be much more conservative in interpretations of their data. The authors have indeed observed phenomena (the presence of CD9 and JUNO in the PVS) that could be consistent with a molecular basis of a means to prevent fertilization by a second sperm. However, the authors would need additional data from additional experimental studies, such as interfering with the release of CD9 and JUNO and showing that this experimental manipulation leads to increased polyspermy, or creating an experimental situation that mimics the presence of CD9 and JUNO (in essence, what the authors call "sperm inhibiting medium" on page 20) and showing that this prevents fertilization.

A major section of the Results section here (starting with "The consequence is that ... ") is

Full Revision

speculation. Rather than be in the Results section, this should be in the Discussion. The language should be also softened regarding the roles of these proteins in the perivitelline space in other portions of the manuscript, such as the abstract and the introduction.

Finally, the authors should do more to discuss their results with the results of Miyado et al. (2008), which interestingly, posited that CD9 is released from the oocytes and that this facilitates fertilization by rendering sperm more fusion-competent. There admittedly are two reports that present data that suggest lack of detection of CD9-containing exosomes from eggs (as proposed by Miyado et al.), but nevertheless, the authors should put their results in context with previous findings.

We generally agree with all the remarks and suggestions made here. In the revised version of the manuscript, we have retained in the *Results* section (pp. 14–15) only the factual data concerning the localization of CD9 and JUNO in unfertilized and fertilized oocytes, as well as in the spermatozoa present in the PVS of these oocytes. We have taken care not to include any interpretive elements in this section, which are now presented exclusively in a dedicated paragraph of the *Discussion*, entitled “Possible molecular bases of the membrane-block and ZP-block contributing to the fusion block” (p. 21). There, we develop our hypothesis and discuss it in light of both the findings from the present study and previous work by other groups. In doing so, we also address the data reported by Miyado et al. (2008, <https://doi.org/10.1073/pnas.0710608105>), as well as subsequent studies by two other groups—Gupta et al. (2009, <https://doi.org/10.1002/mrd.21040>) and Barraud-Lange et al. (2012, <https://doi.org/10.1530/REP-12-0040>)—that have challenged Miyado’s findings.

We are fully aware that our interpretation of the coverage of unfused sperm heads in the perivitelline space (PVS) by CD9 and JUNO, released from the oolemma—as a potential mechanism of sperm neutralization contributing to the PVS block—remains, at this stage, a plausible hypothesis or working model that, as such, warrants further experimental investigation. It is precisely in this spirit that we present it—first in the abstract (p.1), then in the *Discussion* section (p. 21), and subsequently in the *perspective* part of the *Conclusion* section (p. 22).

(3) Many of the authors' conclusions focus on their prior analyses of sperm interaction - beautifully illustrated in Figure 7. However, the authors need to be cautious in their interpretations of these data and generalizing them to mammalian fertilization as a whole, because mouse and other rodent sperm have sperm head morphology that is quite different from most other mammalian species.

In a similar vein, the authors should be cautious in their interpretations regarding the extension of these results to mammalian species other than mouse, given data on numbers of perivitelline sperm (ranging from 100s in some species to virtually none in other species), suggesting that different species rely on different egg-based blocks to polyspermy to varying extents. While these observations of embryos from natural matings are subject to numerous nuances, they

Full Revision

nevertheless suggest that conclusions from mouse might not be able to be extended to all mammalian species.

It is not clear to us whether Reviewer 3's comment implies that we have, at some point in the manuscript, generalized conclusions obtained in mice to other mammalian species—which we have not—or whether it is simply a general, common-sense remark with which we fully agree: that findings established in one species cannot, by default, be assumed to apply to another.

We would like to emphasize that throughout the manuscript, we have taken care to restrict our interpretations and conclusions to the mouse model, and we have avoided any unwarranted extrapolation to other species.

To definitively close this matter—if there is indeed a matter—we have added the following clarifying statements in the revised version of the manuscript:

In the introduction, second paragraph (pp. 2–3): *"The variability across mammalian species in both the rate of fertilized oocytes with additional spermatozoa in their PVS (from 0 to more than 80%) after natural mating and the number of spermatozoa present in the PVS of these oocytes (from 0 to more than a hundred) suggests that the time for completion of the penetration block and thus its efficiency to prevent polyspermy can vary significantly between species."*

At the end of the preamble to the Results section (p. 4): *"This experimental study was conducted in mice, which are the most widely used model for studying fertilization and polyspermy blocks in mammals. While there are many interspecies similarities, the findings presented here should not be directly extrapolated to humans or other mammalian species without species-specific validation."*

In the Conclusion, the first sentence is (p.22) : *"This study sheds new light on the complex mechanisms that enable fertilization and ensure monospermy in mouse model."*

Within the Conclusion section, among the perspectives of this work (p. 22): *"In parallel, comparative studies in other mammalian species will be needed to assess the generality of the PVS-block and its contribution relative to the membrane-block and ZP-blocks, as well as the generality of the mechanical role played by flagellar beating and ZP mechanical constraint in membrane fusion."*

(4) Results, page 4 - It is very valuable that the authors clearly define what they mean by a penetrating spermatozoon and a fertilizing spermatozoon. However, they sometimes appear not to adhere to these definitions in other parts of the manuscript. An example of this is on page 10; the description of penetration of spermatozoon seems to be referring to membrane fusion with the oocyte plasma membrane, which the authors have alternatively called "fertilizing" or fertilization - although this is not entirely clear. The authors should go through all parts of the manuscript very carefully and ensure consistent use of their intended terminology.

Overall, while these definitions on page 4 are valuable, it is still recommended that the authors explicitly state when they are addressing penetration of the ZP and fertilization via fusion of the sperm with the oocyte plasma membrane. This help significantly in comprehension by readers.

Full Revision

An example is the section header in the middle of page 9 - this could be "Spermatozoa can penetrate the ZP after the fertilization, but have very low chances to fertilize."

We chose to define our use of the term penetration at the beginning of the Results section because, as readers of fertilization studies, we have encountered on multiple occasions ambiguity as to whether this term was referring to sperm entry into the perivitelline space following zona pellucida traversal, or to the fusion of the sperm with the oolemma. To avoid such ambiguity, we were particularly careful throughout the writing of our original manuscript to use the term penetration exclusively to describe sperm entry into the PVS. The terms fertilizing and fusion were reserved specifically for membrane fusion between the gametes. However, as occasional lapses are always possible, we followed Reviewer 3's recommendation and carefully re-examined the entire manuscript to ensure consistent use of our intended terminology. We did not identify any inconsistencies, including on page 10, which was cited as an example by Reviewer 3. We therefore confirm that, in accordance with our predefined terminology, all uses of the term penetration, on that page and anywhere else in our original manuscript, refer exclusively to sperm entry into the PVS and do not pertain to fusion with the oolemma.

That said, it is important that all readers—including those who may only consult selected parts of the article—are able to understand it clearly. Therefore, despite the potential risk of slightly overloading the text, Reviewer 3's suggestion to systematically associate the term *penetration* with *ZP* seems to us a sound one. However, we have opted instead to associate *penetration* with *PVS*, as our study focuses on the timing of sperm penetration into the perivitelline space, rather than on the traversal of the zona pellucida itself. Accordingly, except in a few rare instances where ambiguity seemed impossible, we have systematically used the phrasing "*penetration into the PVS*" throughout the revised version of the manuscript.

Another variation of this is in the middle of page 9, where the authors use the terms "fertilization block" and "penetration block." These are not conventional terms, and venture into being jargon, which could leave some readers confused. The authors could clearly define what they mean, particularly with respect to "penetration block,"

This point has already been addressed in our response to Comment 1 from Reviewer 3. We invite Reviewer 3 to refer to that response.

This extends to other portions of the manuscript as well, such as Figure 2C, with the label on the y-axis being "Time after fertilization." It seems that what the authors actually observed here was the cessation of sperm tail motility. (It is not evident they they did an assessment of sperm-oocyte fusion here.)

Regarding **Figure 2C** (original version), it has been merged with **Figure 2B** (original version) to form a single figure (**Figure S2D**), now included in **Supplementary Information S12**. This new figure retains all the information originally presented in Figure 2C and indicates the time axis origin as the time when oscillatory movements of the sperm cease.

Full Revision

That said, for the reasons detailed in our response to Reviewer 1 and in the Materials and Methods, we explain why it is legitimate to use the cessation of sperm head oscillations on the oolemma as a marker for the timing of the fusion event. We invite the reviewers to refer to that response for a full explanation of our rationale.

(5) Several points that the authors try to make with several pieces of data do not come across clearly in the text, including Figure 2 on page 6, Figure 4 on page 9, and the various states utilized for the statistical treatment, "post-first penetration, post-first fertilization, no fertilization, penetration block and polyspermy block" on page 10. Either re-writing and clearer definitions/explanations are needed, and/or schematic illustrations could be considered to augment re-written text. Illustrations could be a valuable way present the intended concepts to readers more clearly and accurately. For example, Figure 4 and the associated text on page 9 get particularly confusing - although this sounds like a quite impressive dataset with observations of 138 sperm. Illustrations could be helpful, in the spirit of "a picture is worth 1000 words," to show what seem to be three different situations of sequences of events with the sperm they observed. Finally, the text in the Results about the 138 sperm is quite difficult to follow. It also might help comprehension to augment the percentages with the actual numbers of sperm - e.g., is 48.6% referring 67 of the total 138 sperm analyzed? Does the 85.1% refer to 57 of these 67 sperm?

Figure 2 in the original version of our manuscript concerns sperm engulfment and PB2 extrusion. As already mentioned in our response to Reviewer 1, the characterization of sperm engulfment and PB2 extrusion kinetics is highly relevant to the analysis of the penetration and fusion blocks. However, we agree that its presence in the main text may distract the reader from the main focus of the study. Therefore, this figure and the associated text have been moved to the Supplementary Information in the revised manuscript (SI 2, pages 26–27).

Regarding Figure 4 (original version), in response to Reviewer 3's concern about the difficulty in grasping the message conveyed in its three graphs and associated text we have completely rethought the way these data are presented. Since the three graphs of Figure 4 were directly derived from the experimental timing data of sperm entry in the PVS and fusion with the oolemma in fertilized oocytes (originally shown in Figure 3A), we have combined them into a single figure in the revised manuscript: **Figure 3 (page 8)**. This **new Figure 3** now comprises three components:

- **Figure 3A** remains unchanged from the original version and shows the timing of sperm penetration and fusion in fertilized oocytes. Each sperm category (fused or non-fused , penetrated in the PVS before fusion or after fusion) is represented using a color code clearly explained in the main text (last paragraph of page 7).
-
- **Figure 3B** focuses specifically on the first spermatozoon to penetrate the PVS of each oocyte. It reports how many of these first-penetrating spermatozoa succeeded in fusing versus how many failed to do so, highlighting that being the first to arrive is not sufficient for fusion—other factors are involved. This is explained simply in the first paragraph of page 9.

Full Revision

- **Figure 3C** considers all spermatozoa that entered the PVS of fertilized oocytes, classifying them into three categories: those that penetrated the PVS before fertilization, those that did so after fertilization, and those for which the timing could not be precisely determined. Such classification makes it apparent that the number of spermatozoa penetrating before and after fertilization is of the same order of magnitude, indicating that fertilization is not very effective at preventing further sperm entry into the PVS for the duration of our observations (~4 hours). To facilitate the identification of these three categories, the same color code used in Figure 3A is applied. In addition, within each category, the number of spermatozoa that successfully fused are indicated in black. This allows the reader to quickly assess the fertilization probability for each category—high for sperm entering before fertilization, very low or null for those entering after fertilization. This analysis shows that fertilization is far more effective at blocking sperm fusion than at blocking sperm penetration. This is clearly explained in the second paragraph of page 9.

Regarding **statistical analysis**, as already mentioned in our responses to Reviewers 1 and 2, this section has been rewritten to improve clarity and readability. The notation has also been significantly simplified. To improve the overall fluidity of the text related to the statistical analysis, **Figure 3B** (original version), which presented the timing of penetration into the perivitelline space of oocytes that remained unfertilized, along with its associated statistical analysis previously in **Figure 5B**, have been revised and transferred together in a single Figure S1 of the Supplementary Information (**S11, pages 26; now Figures S1A and S1B**).

(6) Introduction, page 2 - it is inaccurate to state that only diploid zygotes can develop into a "new being." Triploid zygotes typically fail early in develop, but can survive and, for example, contribute to molar pregnancies. Additionally, it would be beneficial to be more scientifically precise term than saying "development into a new being." This is recommended not only for scientific accuracy, but also due to current debates, including in lay public circles, about what defines "life" or human life.

In response to Reviewer 3's comment, we no longer state in the revised version of the manuscript that only diploid zygotes can develop into a new being. We have modified our wording as follows, on page 2, second paragraph: *"In mammals, oocytes fertilized by more than one spermatozoon cannot develop into viable offspring."*

(7) Introduction, page 2 - The mammalian sperm must pass through three layers, not just two as stated in the first paragraph of the Introduction. The authors should include the cumulus layer in this list of events of fertilization.

The sentence from the introduction from the original manuscript mentioned by Reviewer 3 was: *"To fertilize, a spermatozoon must successively pass two oocyte's barriers."* This statement is accurate in the sense that the cumulus cell layer is not part of the oocyte itself, unlike the two oocyte's barriers: the zona pellucida and the oolemma. Moreover, the traversal of the cumulus layer is not within the scope of our study, unlike the traversal of the zona pellucida and fusion with the oolemma. However, it is also correct that in our study the spermatozoa have passed through

Full Revision

the cumulus layer before reaching the oocyte. Therefore, in response to Reviewer 3's comment, we have revised the sentence to clarify this point as follows:

“Once a spermatozoon has passed through the cumulus cell layer surrounding the oocyte, it still must overcome two oocyte's barriers to complete fertilization.”

(8) Introduction, page 2 - While there is evidence that zinc is released from mouse egg upon fertilization, the evidence is not convincing or conclusive that zinc is released from cortical granules or via cortical granule exocytosis.

To better highlight the rationale, storyline, and scope of our study, the introduction has been thoroughly streamlined. In this context, the section discussing the cortical reaction and zinc release seemed more appropriate in the Discussion, specifically within the paragraph titled “Relationship between the penetration block and the ZP-block.”

To address the uncertainty raised by Reviewer 3 regarding the origin of the zinc spark release, we have rephrased this part as follows:

“The fertilization-triggered processes responsible for the changes in ZP properties are generally attributed to the cortical reaction—a calcium-induced exocytosis of secretory granules (cortical granules) present in the cortex of unfertilized mammalian oocytes—and to zinc sparks. As a result, proteases, glycosidases, lectins, and zinc are released into the perivitelline space (PVS), where they act on the components of the zona pellucida. This leads to a series of modifications collectively referred to as ZP hardening or the ZP-block”.

(9) The authors inaccurately state, "only if monospermic multi-penetrated oocytes are able to develop normally, which to our knowledge has never been proven in mice" (page 4) - This was demonstrated with the Astl knockout, assuming that the authors use of "multi-penetrated oocytes" here refers to the definition of penetration that they use, namely penetrating the ZP. This also is one of the instances where the authors contradict themselves, as they note the results with this knockout on page 18.

Thank you for bringing this point to our attention. Nozawa et al. (2018) found that female mice lacking ovastacin (Astl)—the protease released during the cortical reaction that plays a key role in rendering the zona pellucida impenetrable—are normally fertile. They also reported that oocytes recovered from these females after mating were monospermic, despite the consistent presence of additional spermatozoa in the perivitelline space. We can indeed consider that taken together these findings demonstrate that the presence of multiple spermatozoa in the PVS does not impair normal development, as long as the oocyte remains monospermic. In our study, we re-demonstrated this in a different way (by reimplantation of *monospermic oocytes with additional spermatozoa in their PVS*) in a more physiological context of WT oocytes, but we agree that we cannot state: *“which to our knowledge has never been proven in mice.”* This part of the sentence has therefore been removed. In the revised version of the manuscript, the sentence is now formulated in the first paragraph of page 5 as follows: *“However, the contribution of the fusion*

Full Revision

block to prevent polyspermy has physiological significance only if monospermic oocytes with additional spermatozoa in their PVS can develop into viable pups."

Minor comments:

There are numerous places where this reader marked places of confusion in the text. A sample of some of these:

We will indicate how we have modified the text in the specific examples provided by Reviewer 3. Beyond these, however, we would like to emphasize that we have thoroughly revised the entire manuscript to improve clarity and precision.

Page 4 - "continuously relayed by other if they detach" - don't know what this means

Replaced now p 5 by *"can be replaced by others if they detach"*

Page 6 - "hernia" - do the authors mean "protrusion" on the oocyte surface?

The paragraph from the Results section in question has now been moved to the Supplementary Information, on pages 26 and 27. The term *hernia* has been systematically replaced with *protrusion*, including in the Materials and Methods section on page 24.

Page 10 - "penetration of spermatozoa in the PVS falls down" - don't know what this means

Falls down has been removed from the new version and replaced with decreases

Page 12 - "spermatozoa linked to the oocyte ZP" - not clear what "linked" means here

Replaced now page 16 by *"spermatozoa bound to the oocyte ZP"*

Page 14 - "by dint of oscillations" - don't know what this means

Replaced now page 10 by *"the persistent flagellum movements"*

Specifics for Materials and Methods:

Exact timing of females receiving hCG and then being put with males for mating - assume this was immediate but this is an important detail regarding the timing for the creation of embryos in vivo.

That is correct: females were placed with males for mating immediately after receiving hCG. This clarification has been added in the revised version of the manuscript.

Full Revision

Please provide the volumes in which inseminations occurred, and how many eggs were placed in this volume with the 10^6 sperm/ml.

The number of eggs may vary from one cumulus–oocyte complex to another. It is therefore not possible to specify exactly how many eggs were inseminated. However, we now indicate on page 23 the number of cumulus–oocyte complexes inseminated (4 per experiment), the volume in which insemination was performed (200 μ L), and the sperm concentration used 10^6 sperm/mL.

****Referees cross-commenting****

I concur with Reviewer 1's comment, that the 'challenging prior dogma' about the first sperm not always being the one to fertilize the egg is too strong. As Reviewer 1 notes, "it had been observed before that it is not necessarily the first sperm that gets through the ZP that fertilizes the egg." I even thought about adding this comment to my review, although held off (I was hoping to find references, but that was taking too long).

Please refer to our response to Reviewer 1 regarding this point.

Dear Dr. Gourier

Thank you for the submission of your revised manuscript to our journal. Since my colleague Esther Schnapp is currently traveling, I have temporarily taken over the handling of your manuscript.

It has been seen again by former referee #1 who considers your response to all concerns adequate and supports publication after minor textual changes. Please provide a point-by-point response to the remaining concerns.

Based on this positive evaluation, I invite you to revise your study for publication in EMBO reports.

The revision will comprise two steps:

- 1) Please follow the general formatting requests listed below (section B). I also added a few specific requests under (A).
- 2) Shortly after I invited revision you will receive a separate email with instructions for providing source data with your revised manuscript, including how to upload and organize the files.
- 3) Once the revision is ready, please upload it. We will then, with all files and source data at hand, perform a few more editorial checks on statistics, image integrity etc. All further formatting requests can then be implemented before formal acceptance.

=====

A) A few SPECIFIC FORMATTING REQUESTS (some are also listed in the general formatting part with more information):

- 1) Please write the abstract in present tense and keep the word limit in mind (175 words).
- 2) Please submit all figures as individual production quality files (see also below).
- 3) For all quantifications, you need to show individual datapoints in addition to the mean/median and error bars (e.g., Figure 2B, C, Figure 7D). We need the exact p-values (unless $p < 0.0001$).
- 4) The nomenclature for "Video" is "Movie EV#". The movies are zipped together with their legend (README.txt file) and the zipped folder is uploaded.
- 5) The scale bar size must be defined in the figure legend, not in the figure itself.
- 6) The references should be alphabetical. 'et al' is used after 10 authors, if there are more than that.
- 7) We do not have a paragraph called "Conclusion". Please either incorporate it in the Discussion or remove it, as you see best fit.
- 8) Supplementary information is either part of a PDF called "Appendix" or uploaded as Expanded View content (Figure EV#, EV Table # etc). Your Figures S1 and S2 could be Figure EV1 and EV2. Their legends are part of the manuscript in a section called Expanded View Figure legends.
- 9) The ethics statement should be part of the Methods section.
- 10) Author contributions need to be removed from the manuscript and solely be defined in the online manuscript tracking system. Our production team will typeset this information into the article.
- 11) We need a Reagents and Tools Table and an Author checklist.
- 12) The manuscript must contain a Data Availability section (see general point 7)

=====

B) GENERAL FORMATTING INSTRUCTIONS:

- 1) a .docx formatted version of the manuscript text (including legends for main figures, EV figures and tables). Please make sure that the changes are highlighted to be clearly visible.
- 2) individual production quality figure files as .eps, .tif, .jpg (one file per figure). Please download our Figure Preparation Guidelines (figure preparation pdf) from our Author Guidelines pages <https://www.embopress.org/page/journal/14693178/authorguide> for more info on how to prepare your figures.
- 3) a .docx formatted letter INCLUDING the reviewers' reports and your detailed point-by-point responses to their comments. As part of the EMBO Press transparent editorial process, the point-by-point response is part of the Review Process File (RPF), which will be published alongside your paper.
- 4) a complete author checklist, which you can download from our author guidelines (). Please insert information in the checklist that is also reflected in the manuscript. The completed author checklist will also be part of the RPF.
- 5) Please note that all corresponding authors are required to supply an ORCID ID for their name upon submission of a revised

manuscript (). Please find instructions on how to link your ORCID ID to your account in our manuscript tracking system in our Author guidelines
()

6) We replaced Supplementary Information with Expanded View (EV) Figures and Tables that are collapsible/expandable online. A maximum of 5 EV Figures can be typeset. EV Figures should be cited as 'Figure EV1, Figure EV2' etc... in the text and their respective legends should be included in the main text after the legends of regular figures.

7) Please include a dedicated "Data Availability" section at the end of the Methods (suggested wording: "The [structural coordinates | microarray | mass spectrometry] data from this publication have been deposited to the [name of the database] database [URL] and assigned the identifier [accession | permalink | hashtag]."). Should this not apply, this should still be stated as "This study includes no data deposited in external repositories."

Additional information on source data and instruction on how to label the files are available

10) Figure legends and data quantification:

- the name of the statistical test used to generate error bars and P values,
- the EXACT p-values,
- the number (n) of independent experiments (please specify technical or biological replicates) underlying each data point,
- the nature of the bars and error bars (s.d., s.e.m.)

- If the data are obtained from n {less than or equal to} 5, show the individual data points in addition to the SD or SEM.

- If the data are obtained from n {less than or equal to} 2, use scatter blots showing the individual data points.

11) Our journal encourages inclusion of *data citations in the reference list* to directly cite datasets that were re-used and obtained from public databases. Data citations in the article text are distinct from normal bibliographical citations and should directly link to the database records from which the data can be accessed. In the main text, data citations are formatted as follows: "Data ref: Smith et al, 2001" or "Data ref: NCBI Sequence Read Archive PRJNA342805, 2017". In the Reference list, data citations must be labeled with "[DATASET]". A data reference must provide the database name, accession number/identifiers and a resolvable link to the landing page from which the data can be accessed at the end of the reference. Further instructions are available at .

12) All Materials and Methods need to be described in the main text using our 'Structured Methods' format. According to this format, the Methods section includes a Reagents and Tools Table (listing key reagents, experimental models, software and relevant equipment and including their sources and relevant identifiers) followed by a Methods and Protocols section describing the methods, ideally using a step-by-step protocol format. The aim is to facilitate adoption of the methodologies across labs. Please download and fill our Reagents and Tools Table template (.docx), which you can find in our author guidelines: <https://www.embopress.org/page/journal/14693178/authorguide#structuredmethods>.

<https://www.embo.org/doi/10.15252/msb.20178071>.

13) As part of the EMBO publication's Transparent Editorial Process, EMBO Reports publishes online a Review Process File to accompany accepted manuscripts. This File will be published in conjunction with your paper and will include the referee reports, your point-by-point response and all pertinent correspondence relating to the manuscript.

Yours sincerely,

=====

Referee #1:

I have read the revised manuscript as well as the comments of the authors in response to the points I raised. Thank you for providing overall clear and well-explained reasonings to your statements and results, that helped in understanding and clarifying most of the points I had. Overall the authors have improved the readability of the manuscript greatly by adding schemes and being more precise in the wording.

Regarding the Hoechst-based fusion assay: Thank you for explaining that the conventional 'fusion assay' (Hoechst incubation of oocytes to see transfer of Hoechst to fusing sperm) is technically not possible via incubating cumulus-cells containing oocytes. I don't know whether injecting the Hoechst dye into the oocyte would be feasible as alternative, but the additional explanation why the cessation of sperm movement can serve as proxy for sperm-oocyte fusion is ok, too.

Regarding providing a staining of CD9 and JUNO levels on PVS-sperm pre-fusion: While I understand that the authors had only imaged 4 hours post insemination oocytes for their original manuscript and had thus only obtained Juno/CD9 staining for those later time-points, I don't understand why the authors cannot just fix and stain oocytes earlier post insemination, ideally at a time when the chance that sperm has fused already is very low but where they have observed in most cases sperm to be already in the PVS. If this is too variable between oocytes, one could even imagine observing live 1-2 sperm in the PVS but none of them yet fused and then immediately fixing the sample for IF. That should in principle be technically not too difficult to do, compared to the other much more difficult live-cell imaging approaches. However, I don't think it will provide such a clear answer after having seen the result that even in unfertilized oocytes after 4 hours there is still coating with CD9 and JUNO of the PVS-sperm - meaning the release appears to happen even independently of fusion. It might even be more of an internal timer that constantly 'ticks' but is switched to releasing at much higher rate upon fusion. So while I still think it would strengthen the suggestive point of the last sentence of the abstract if the authors had a true 'pre-fusion' staining (and could show that there the coating is much lower for sperm in the PVS), together with prior data that shows that JUNO is released after fusion I do think the working model the authors propose is sufficiently strongly supported by data.

Last point: Please revise the last sentence in the abstract - this is currently really hard to read and understand. Here a suggestion:

We observed that non-fertilizing spermatozoa in the perivitelline space become coated with CD9 and JUNO, oocyte proteins abundantly released from the oolemma after gamete fusion. This finding supports the hypothesis that the post-fusion block to polyspermy involves a perivitelline space mediated mechanism, where supernumerary sperm are neutralized by these and potentially other oocyte derived molecules.

Referee #1:

I have read the revised manuscript as well as the comments of the authors in response to the points I raised. Thank you for providing overall clear and well-explained reasonings to your statements and results, that helped in understanding and clarifying most of the points I had. Overall the authors have improved the readability of the manuscript greatly by adding schemes and being more precise in the wording.

Regarding the Hoechst-based fusion assay: Thank you for explaining that the conventional 'fusion assay' (Hoechst incubation of oocytes to see transfer of Hoechst to fusing sperm) is technically not possible via incubating cumulus-cells containing oocytes. I don't know whether injecting the Hoechst dye into the oocyte would be feasible as alternative, but the additional explanation why the cessation of sperm movement can serve as proxy for sperm-oocyte fusion is ok, too.

Regarding providing a staining of CD9 and JUNO levels on PVS-sperm pre-fusion: While I understand that the authors had only imaged 4 hours post insemination oocytes for their original manuscript and had thus only obtained Juno/CD9 staining for those later time-points, I don't understand why the authors cannot just fix and stain oocytes earlier post insemination, ideally at a time when the chance that sperm has fused already is very low but where they have observed in most cases sperm to be already in the PVS. If this is too variable between oocytes, one could even imagine observing live 1-2 sperm in the PVS but none of them yet fused and then immediately fixing the sample for IF. That should in principle be technically not too difficult to do, compared to the other much more difficult live-cell imaging approaches. However, I don't think it will provide such a clear answer after having seen the result that even in unfertilized oocytes after 4 hours there is still coating with CD9 and JUNO of the PVS-sperm - meaning the release appears to happen even independently of fusion. It might even be more of an internal timer that constantly 'ticks' but is switched to releasing at much higher rate upon fusion. So while I still think it would strengthen the suggestive point of the last sentence of the abstract if the authors had a true 'pre-fusion' staining (and could show that there the coating is much lower for sperm in the PVS), together with prior data that shows that JUNO is released after fusion I do think the working model the authors propose is sufficiently strongly supported by data.

Last point: Please revise the last sentence in the abstract - this is currently really hard to read and understand. Here a suggestion:

We observed that non-fertilizing spermatozoa in the perivitelline space become coated with CD9 and JUNO, oocyte proteins abundantly released from the oolemma after gamete fusion. This finding supports the hypothesis that the post-fusion block to polyspermy involves a perivitelline space mediated mechanism, where supernumerary sperm are neutralized by these and potentially other oocyte derived molecules.

Rev_Com_number: RC-2024-02719

New_manu_number: EMBOR-2025-62296V1-T

Corr_author: Gourier

Title: Key roles of the zona pellucida in promoting gamete fusion and fusion block to polyspermy in mice

=====

Referee #1:

I have read the revised manuscript as well as the comments of the authors in response to the points I raised. Thank you for providing overall clear and well-explained reasonings to your statements and results, that helped in understanding and clarifying most of the points I had. Overall the authors have improved the readability of the manuscript greatly by adding schemes and being more precise in the wording.

Regarding the Hoechst-based fusion assay: Thank you for explaining that the conventional 'fusion assay' (Hoechst incubation of oocytes to see transfer of Hoechst to fusing sperm) is technically not possible via incubating cumulus-cells containing oocytes. I don't know whether injecting the Hoechst dye into the oocyte would be feasible as alternative, but the additional explanation why the cessation of sperm movement can serve as proxy for sperm-oocyte fusion is ok, too.

Regarding providing a staining of CD9 and JUNO levels on PVS-sperm pre-fusion: While I understand that the authors had only imaged 4 hours post insemination oocytes for their original manuscript and had thus only obtained Juno/CD9 staining for those later time-points, I don't understand why the authors cannot just fix and stain oocytes earlier post insemination, ideally at a time when the chance that sperm has fused already is very low but where they have observed in most cases sperm to be already in the PVS. If this is too variable between oocytes, one could even imagine observing live 1-2 sperm in the PVS but none of them yet fused and then immediately fixing the sample for IF. That should in principle be technically not too difficult to do, compared to the other much more difficult live-cell imaging approaches. However, I don't think it will provide such a clear answer after having seen the result that even in unfertilized oocytes after 4 hours there is still coating with CD9 and JUNO of the PVS-sperm - meaning the release appears to happen even independently of fusion. It might even be more of an internal timer that constantly 'ticks' but is switched to releasing at much higher rate upon fusion. So while I still think it would strengthen the suggestive point of the last sentence of the abstract if the authors had a true 'pre-fusion' staining (and could show that there the coating is much lower for sperm in the PVS), together with prior data that shows that JUNO is released after fusion I do think the working model the authors propose is sufficiently strongly supported by data.

We thank Reviewer 1 for the constructive review of our manuscript. The reflection provided on the kinetics of CD9 and JUNO levels on PVS-sperm pre and post-fusion is highly relevant and represents an independent line of investigation that will be valuable to explore in a future study.

Last point: Please revise the last sentence in the abstract - this is currently really hard to read and understand. Here a suggestion:

We observed that non-fertilizing spermatozoa in the perivitelline space become coated with CD9 and JUNO, oocyte proteins abundantly released from the oolemma after gamete fusion. This finding supports the hypothesis that the post-fusion block to polyspermy involves a perivitelline space mediated mechanism, where supernumerary sperm are neutralized by these and potentially other oocyte derived molecules.

Since the Abstract must not exceed 175 words, we had to rewrite it entirely. Therefore, we could not retain the wording proposed by Reviewer 1 in its entirety, but we drew inspiration from it to make the final sentence clearer and easier to read.

Dear Dr. Gourier,

Thank you for the submission of your revised manuscript. As my colleague Martina already mentioned, we performed our final checks now and a few minor changes are still required before we can proceed with the official acceptance:

- The conflict of interest subheading needs to be corrected to Disclosure and Competing Interests Statement.
- The author CHECKLIST has not been fully completed. You also need to chose responses in cells D113, D114, D115 please.
- The following figure callouts need to be corrected: Figure S2A and S2C, Figure S2C, Figures S2C and S2D, they are all missing the word Appendix. Appendix Fig S2A, etc. is correct.
- In the APPENDIX FILE each item (figure) needs to be listed in the table of content on the title page as its full name: Appendix Figure S1 and Appendix Figure S2; the titles of the figures in the legends need to be corrected to Appendix Figure S1 and Appendix Figure S2.
- 5 movies are uploaded. The movies have been zipped up in one folder, but we need each legend (in txt) and each movie file zipped up together, so that we have 5 separate zip folders uploaded called Movie EV1, Movie EV2, etc.
- The SYNOPSIS IMAGE is very busy and the file is too large. We need an image file that is 550 pixels wide x 200-600 pixels high and all text needs to be readable at the final image size. Please send us a new image, ideally one that is less busy.
- We also need a SYNOPSIS TEXT: A short (1-2 sentences) summary of the findings and their significance, and 2-3 bullet points highlighting key results.
- The Reagents & Tools table in the ms needs to be removed. We only need the table uploaded as a separate file.
- The Figure legends need to be listed after the References, at the very end of the manuscript file.
- We received a bounced email for co-author Nathan Martin-Fornier - nathan.mafo@proton.fr; please either remove the author from the author list and add him back using the new email address or send us the new email and we will update the account accordingly.
- Please note that the exact p values are not provided in the legend of figure 2b. Please provide exact values as reasonable.
- Please note that the scale bar is missing for figure 7a.

I would like to suggest a minor change to the title. Do you agree with :

Roles of the zona pellucida in gamete fusion and of the perivitelline space in blocking polyspermy in mice

I also would like to suggest to delete the word "found" from the last sentence in the abstract. Please let me know whether you agree with these changes.

When you start a new version of your ms in our online submission system, you can bring forward the old files and then replace only the files that need to be replaced, which hopefully makes it easier.

The authors have addressed all minor editorial requests.

Dr. Christine Gourier
Laboratoire de Physique de l'Ecole Normale Supérieure de Paris
Physics department
24 rue Lhomond
Paris 75005
France

Dear Dr. Gourier,

I am very pleased to accept your manuscript for publication in the next available issue of EMBO reports. Thank you for your contribution to our journal.

Yours sincerely,
